# DTS: Enhancing Large Reasoning Models via Decoding Tree Sketching

Zicheng Xu [* 1]  Xiuyi Lou [* 1]  Guanchu Wang [† * 2]  Yu-Neng Chuang [3]  Feng Luo [3]  Guangyao Zheng [1]
Alexander S. Szalay [1]  Zirui Liu [4]  Vladimir Braverman [† 1]

## Abstract

Large Reasoning Models (LRMs) achieve remarkable inference-time improvements through parallel thinking. However, existing methods rely on redundant sampling of reasoning trajectories, failing to effectively explore the reasoning space to uncover high-quality solutions. To address these limitations, we propose Decoding Tree Sketching (`DTS`), a plug-and-play decoding framework for structural multi-trajectory exploration and reasoning selection. For reasoning exploration, `DTS` sketches a backbone tree of the reasoning space by selectively branching at decision tokens. For reasoning selection, guided by length-accuracy anti-correlation, `DTS` designs an early termination to prioritize short and reliable trajectories during decoding. Experimental results across four LRMs and datasets demonstrate that `DTS` significantly enhances accuracy by **14%** and reduces repetitive generation by **8%** on average. Notably, `DTS` enables smaller models to outperform larger models with 10× the size, highlighting its potential to strengthen reasoning capabilities. The source code is available at: https://github.com/ZichengXu/Decoding-Tree-Sketching, with a vLLM implementation at: https://github.com/meierteng/DTS-vLLM.

[1]Department of Computer Science, Johns Hopkins University, Baltimore, USA [2]Department of Computer Science, University of North Carolina at Charlotte, Charlotte, USA [3]Department of Computer Science, Rice University, Houston, USA [4]Department of Computer Science, University of Minnesota, Minneapolis, USA. Correspondence to: Vladimir Braverman <vova@cs.jhu.edu>, Guanchu Wang <gwang16@charlotte.edu>.

*Proceedings of the 43rd International Conference on Machine Learning*, Seoul, South Korea. PMLR 306, 2026. Copyright 2026 by the author(s).

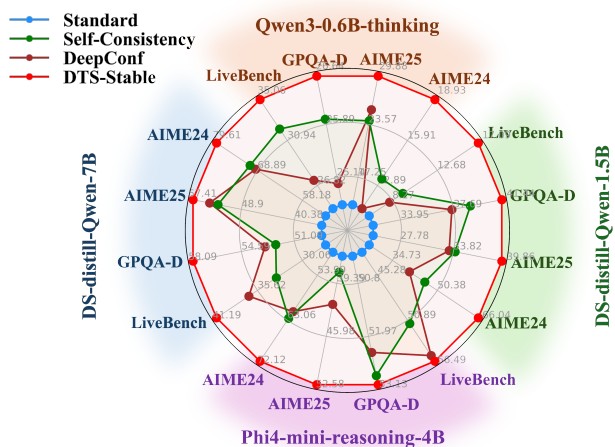

*Figure 1.* Comprehensive performance of `DTS` on the AIME24, AIME25, GPQA-D, LiveBench datasets.

## 1. Introduction

Large Reasoning Models (LRMs), such as DeepSeek-R1 (DeepSeek-AI, 2025) and Qwen3 (Team, 2025), have demonstrated impressive reasoning capabilities across domains such as mathematics, programming, and scientific problem-solving (Zhang et al., 2025; Chuang et al., 2025; Xu et al., 2025). These models have shown strong inference-time reasoning performance on challenging tasks (Leang et al., 2025a; Zhou et al., 2024) by generating explicit, step-by-step chains of thought (CoT) (Wei et al., 2022; Neeley et al., 2025). Beyond single reasoning trajectory, recent work has shown that reasoning through parallel thinking can substantially enhance LRM performance. Methods such as Self-Consistency (Wang et al., 2023b) and DeepConf (Fu et al., 2025) generate multiple reasoning trajectories and aggregate their results, allowing models to offset local errors and consider alternative solutions. Empirically, such multi-trajectory decoding often outperforms single-trajectory inference, highlighting parallel thinking as a powerful mechanism for improving reasoning accuracy.

However, existing parallel thinking approaches exhibit two fundamental limitations. First, most methods independently

generate reasoning trajectories. The parallel reasoning processes are often highly identical and redundant, failing to provide meaningful semantic exploration of the underlying reasoning space. As a result, increasing the number of samples does not necessarily lead to better coverage or diversity of reasoning trajectories (Zhang et al., 2026). Second, trajectory selection is typically applied only post hoc, after all reasoning sequences have fully completed, providing insufficient robust principles to guide or prioritize promising reasoning trajectories during decoding. Together, these limitations suggest that effective parallel thinking requires both **structured exploration** and **reliable trajectory selection**. To address these limitations, we propose **D**ecoding **T**ree **S**ketching (DTS) that sketches a structured reasoning tree and prioritizes high-quality solutions via length-aware early termination within a unified decoding framework.

### How to structurally explore the reasoning space?

Exploring multiple reasoning trajectories can be viewed as searching over an implicit reasoning space. LRM stochastic decoding induces a space of alternative reasoning trajectories that can be naturally organized as a tree structure, where each node corresponds to a generated token and each root-to-leaf path represents a complete CoT reasoning trajectory. Conceptually, an oracle that could exhaustively enumerate and evaluate all paths in this tree would identify the best reasoning trajectory and recover the correct answer. However, in practice, the exponential growth of this tree produces a practically infinite search space, rendering exhaustive enumeration computationally infeasible. Therefore, our proposed DTS **sketches** the reasoning space into a compact backbone tree that preserves critical branching structure while avoiding full enumeration. DTS adopts next-token varentropy and entropy to identify decision tokens where several semantically plausible continuations exist. By selectively expanding branches at these tokens, a dynamically sketched reasoning tree is constructed, enabling structural exploration of the reasoning space.

### Is there any insight to support the selection?

Our analysis in Section 2.2 reveals an **anti-correlation** between reasoning length and task accuracy. Empirically, we observe that high-quality solutions often lie along relatively short trajectories within the reasoning tree. We also provide a theoretical foundation for this anti-correlation by examining the Group Relative Policy Optimization (GRPO) (Shao et al., 2024) training objective, which introduces an asymmetric bias that favors short successful reasoning and long unsuccessful ones. Building upon this insight, DTS proposes an early termination criteria that prioritizes the earliest completed trajectories for the final solution. This allows promising solutions to be identified directly during decoding, aligning trajectory selection with the anti-correlation to favor short, reliable trajectories.

### How does DTS perform in practice?

We evaluate DTS on four LRMs across four reasoning datasets. As demonstrated in Figure 1, DTS attains accuracy improvements of **14%** on average. Our contributions are summarized as follows:

- **Training-free decoding:** DTS requires no training, operates entirely at decoding time, and serves as a plug-and-play module for general LRMs.

- **Structured tree sketching:** DTS constructs a dynamic reasoning tree by selectively branching at decision tokens, enabling the exploration of multiple diverse reasoning trajectories within a structured search space.

- **Reliable selection:** DTS leverages the anti-correlation between accuracy and reasoning length to select short yet reliable trajectories for final solution.

- **Evaluation:** Across four LRMs and datasets, DTS consistently improves accuracy and reduces repetition.

## 2. Preliminary

### 2.1. Notations

We consider LRM $f$ in this work. For each input prompt $x$, the reasoning trajectory is auto-regressively generated by $\xi_{t+1} = \xi_t \oplus v_t$, and $v_t \sim f(x_t, \xi_t)$ for $t = 1, 2, \cdots$, where $\xi_t \oplus v_j$ denotes to attach token $v_j$ to trajectory $\xi_t$.

### 2.2. Length-Accuracy Anti-correlation

In this section, we reveal a **length-accuracy anti-correlation** for LRMs, where shorter reasoning trajectories consistently achieve higher task accuracy than longer ones.

**Empirical Evidence.** We comprehensively validate the length-accuracy anti-correlation on the AIME24 (Maxwell-Jia, 2025), AIME25 (math ai, 2025), and GPQA-D (Rein et al., 2023) datasets using DeepSeek-Distill-Qwen-1.5B, -7B (DeepSeek-AI, 2025), and Qwen3-0.6B (Team, 2025). For each problem, we sample 100 stochastic reasoning trajectories. We partition these trajectories into groups based on their length and calculated the mean accuracy for each group. Figure 2 illustrates this relationship, where each point represents the average accuracy for a specific group. Across all cases, we observe a clear anti-correlation with accuracy decreases as length increases, indicating that shorter reasoning chains are more reliable. This length-accuracy anti-correlation motivates DTS's objective to select the shortest reasoning trajectories in the reasoning space and enhance LRMs' performance. Beyond its empirical validation, this anti-correlation is consistent with the Reinforcement Learning (RL) objectives used to train LRMs.

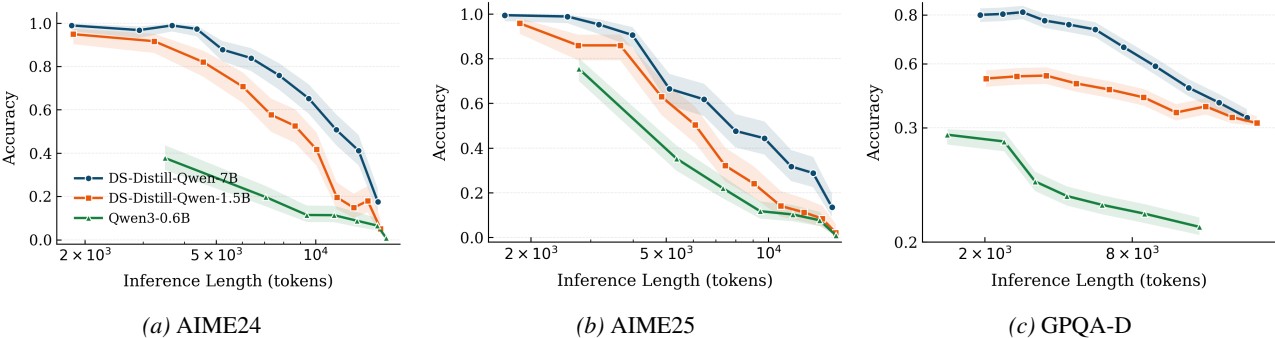

*Figure 2.* Demonstration of the anti-correlation between accuracy and reasoning length. Experiments are conducted with the DS-Distill-Qwen-1.5B, 7B and Qwen3-0.6B LRMs on the (a) AIME24, (b) AIME25, and (c) GPQA-D datsets.

**Theoretical Foundation.** Recent studies have shown that RL post-training introduces a systematic length bias in LRMs (Liu et al., 2025; Devic et al., 2025). Under GRPO objectives, each generated reasoning trajectory is optimized by aggregating token-level updates weighted by a normalized advantage. The abbreviated GRPO objective[1] can be written as

$$\mathcal{J}_{GRPO}(f) = \mathbb{E}_{\mathcal{G} \sim f} \left[ \frac{1}{|\mathcal{G}|} \sum_{\xi \in \mathcal{G}} \frac{1}{|\xi|} \sum_{t=1}^{|\xi|} \ell_t \hat{A}_t \right], \quad (1)$$

where $\mathcal{G}$ is the group of sampled trajectories, $\ell_t$ is the clipped PPO term, and $\hat{A}_t$ is the normalized advantage assigned to the trajectory $\xi$. The length normalization by $|\xi|$ in the equation induces an asymmetric learning bias. For correct reasoning with a positive advantage ($\hat{A}_t > 0$), shorter trajectories receive larger effective updates. Conversely, for incorrect reasoning with a negative advantage ($\hat{A}_t < 0$), longer trajectories reduce the penalty. Therefore, this bias implicitly motivates LRMs to correlate correctness with brevity and incorrectness with verbosity, providing a theoretical foundation for an anti-correlation between reasoning length and accuracy in LRMs.

### 2.3. Chasing Shortest Reasoning Trajectories by Decoding Tree

We follow the anti-correlation between the accuracy and reasoning length to optimize the reasoning process. To represent the reasoning space, all possible reasoning trajectories of an LRM can be naturally represented as a tree structure, where each node corresponds to a possible token in the generated trajectory. Starting from the first token, every step in the reasoning process branches into $|\mathbb{T}|$ possible continuations, where $\mathbb{T}$ denotes the token space. In the following steps, the second token branches into $|\mathbb{T}|^2$ possible paths, the third token into $|\mathbb{T}|^3$, and so forth, leading to an

---

[1]The full GRPO objective can be referred to Equation (1) in the DeepSeek-R1 technical report (DeepSeek-AI, 2025).

exponentially growing tree space $|\mathbb{T}| + |\mathbb{T}|^2 + |\mathbb{T}|^3 + \cdots$.

According to the anti-correlation, shorter reasoning trajectories generally achieve higher accuracy, suggesting that the **optimal solution lies in identifying the shortest paths from the root token to a leaf node**. However, the exponential explosion of the reasoning tree produces an effectively infinite search space, making it computationally infeasible to exhaustively traverse every path for finding the globally optimal reasoning trajectory. To this end, our DTS can effectively prune the search space by sketching the growing tree space during decoding, and approximates the global optimal solution by selecting the shortest reasoning trajectories, rather than exhaustive full enumeration.

## 3. Decoding Tree Sketching (DTS)

We introduce DTS to sketch the decoding tree and approximate the optimal solution. The overall framework of DTS is illustrated in Figure 3. DTS consists of two key modules: decoding tree generation and early termination strategy. Together, these modules enable structured exploration of the reasoning tree and reliable trajectory selection, allowing DTS to select the optimal reasoning path.

### 3.1. Decoding Tree Generation

DTS constructs the decoding tree with two key components: selective branch generation and parallel branch expansion. These components define how the tree grows and decodes trajectories simultaneously. We describe each component in detail below.

**Selective Branch Generation.** Unlike standard autoregressive decoding, which generates a single token at each step, DTS adaptively expands branches when a decision token is encountered, where several semantically distinct continuations remain plausible. Specifically, a decision token's next-token distribution should have several preferred tokens, each carrying noticeable probability mass and cor-

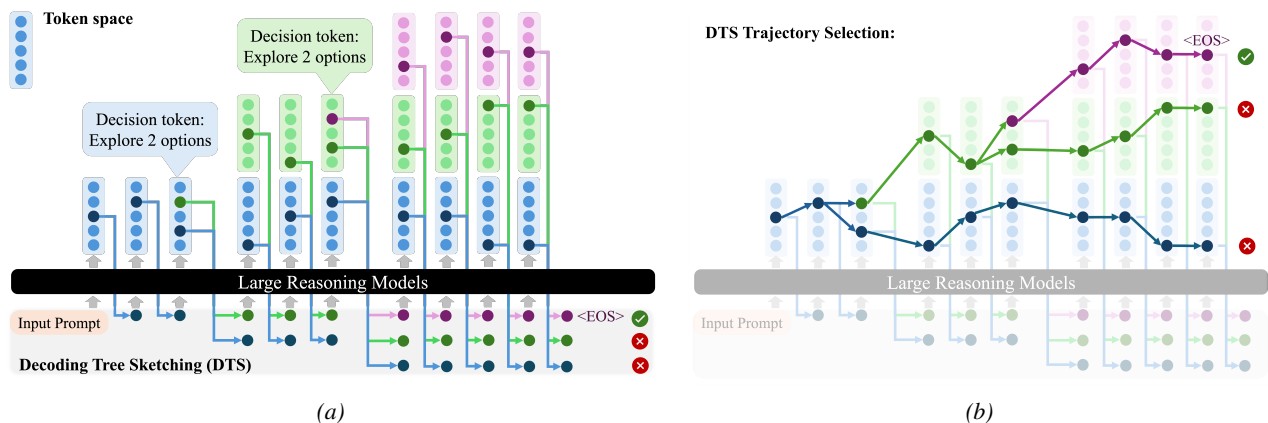

*Figure 3.* (a) Generation of the decoding tree by `DTS`. Starting from the input prompt, `DTS` first follows standard auto-regressive decoding, producing a single branch (blue). At decision tokens, `DTS` generates new branches (green and purple) by selecting top-$K$ candidate tokens. Each branch expands in parallel across its own token space. (b) `DTS` trajectory selection via early termination. All branches share prefix computation up to their branching points. `DTS` prioritizes the earliest completed branches that reach the end-of-sequence token (purple).

responding to different reasoning directions. To identify these decision tokens, `DTS` examines entropy and varentropy of the next-token distribution. Entropy reflects the concentration of probability mass and decreases when the distribution is highly peaked at a few high-probability tokens. Varentropy measures the variance of the information content and increases when the distribution exhibits a high disparity in the surprisals of its top candidates, signaling a competition between different reasoning directions. Therefore, decision tokens are characterized by low entropy and high varentropy where the model commits strongly to a small yet semantically distinct set of continuations, making branching especially valuable.

Formally, given an input prompt $x$ and an intermediate reasoning trajectory $\xi_t$, let $P_t(v) = f(x, \xi_t)$ denote the next-token distribution produced by the LRM at step $t$. We first define a decision token criteria $\delta_{\text{dec}}(P_t)$ that determines whether a token is a decision token:

$$\delta_{\text{dec}}(P_t) = \begin{cases} 1 & \text{if } \text{VE}(P_t) \geq \tau_{\text{v}} \text{ and } \text{H}(P_t) \leq \tau_{\text{h}}, \\ 0 & \text{otherwise,} \end{cases} \quad (2)$$

where $\text{H}(P_t) = -\sum_v P_t(v) \log P_t(v)$ is the entropy, $\text{VE}(P_t) = \sum_v P_t(v)\big(-\log P_t(v) - \text{H}(P_t)\big)^2$ is the varentropy, and $\tau_{\text{v}}$ and $\tau_{\text{h}}$ are the thresholds of varentropy and entropy to control the tradeoff between decoding exploration and computational cost.

The branching function $F(x, \xi_t)$ is then defined as

$$F(x, \xi_t) = \begin{cases} \{v_1, \ldots, v_K\} & \text{if } \delta_{\text{dec}}(P_t) = 1, \\ \{v_1\}, \ v_1 \sim P_t & \text{otherwise,} \end{cases} \quad (3)$$

where $\{v_1, \ldots, v_K\}$ denotes the top-$K$ probable tokens under $P_t$. Rather than branching at every step, `DTS` selectively

generates new branches only at decision tokens ($\delta_{\text{dec}}(P_t) = 1$), where the model assigns substantial probability mass to a small set of alternative continuations. In all other cases, when the prediction is either clearly deterministic or broadly uncertain ($\delta_{\text{dec}}(P_t) = 0$), `DTS` refrains from unnecessary branching and conserves space. In the extreme case $\tau_{\text{v}} \to +\infty$, `DTS` reduces to standard auto-regressive decoding with a single token generated at each step.

**Parallel Branch Expansion.** `DTS` performs auto-regressive generation across all branches in parallel, as illustrated in Figure 3 (a). At each time step $t$, `DTS` maintains a batch of reasoning trajectories $\mathcal{T}_t = \{\xi_t^1, \xi_t^2, \ldots\}$, initialized with $\mathcal{T}_0 = \varnothing$. For every trajectory $\xi_t^i \in \mathcal{T}_t$, the model generates the next tokens based on the branch function $F(x, \xi_t^i)$ from Equation (3), and attaches them to form new trajectories $\{\xi_t^i \oplus v_j \mid v_j \in F(x, \xi_t^i)\}$. Consequently, the reasoning set is updated as

$$\mathcal{T}_{t+1} = \{\xi_t^i \oplus v_j \mid v_j \in F(x, \xi_t^i), \xi_t^i \in \mathcal{T}_t\}. \quad (4)$$

This process iterates for $t = 0, 1, 2, \ldots$, progressively expanding all branches following Equation (4).

### 3.2. Early Termination Strategy

The early termination strategy in `DTS` is motivated by the anti-correlation in Figure 2, where the short reasoning process consistently outperforms the long reasoning process. Specifically, `DTS` provides two termination strategies: `DTS-Greedy` and `DTS-Stable`.

**DTS-Greedy.** `DTS-Greedy` follows a greedy stopping rule, terminating as soon as any candidate trajectory completes the generation. At each decoding step, `DTS` monitors all active reasoning trajectories within the decoding tree

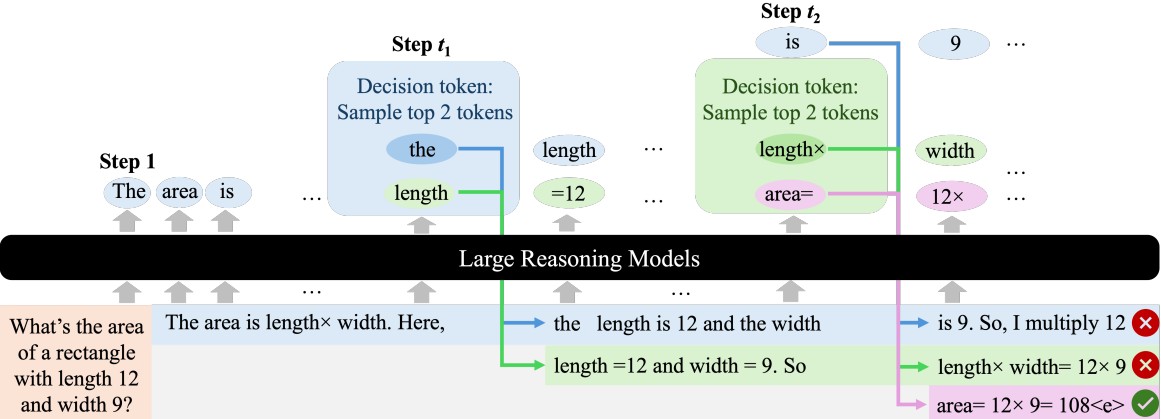

**Figure 4.** An example of `DTS-Greedy` decoding process with $K = 2$, given the input prompt 'What's the area of a rectangle with length 12 and width 9?'. `DTS` selectively generates new branches at steps $t_1$ and $t_2$, and terminates as soon as any branch terminates with an ending token. The final output is 'The area is length× width. Here, length =12 and width = 9. So area= 12× 9= 108'.

to identify if any have generated the end-of-sequence `EOS` token. If at least one trajectory concludes, the generation process terminates immediately and this completed trajectory is returned as the final reasoning trace and answer.

**DTS-Stable.** `DTS-Stable` relaxes the greedy behavior in order to obtain more robust decisions. Instead of immediately committing to the first terminating trajectory, `DTS-Stable` collects a voting budget of $B$ trajectories that have finished and employs majority voting to select a final answer among them. In this way, `DTS-Stable` preserves the early-completion principle motivated by the length-accuracy anti-correlation, while yielding decisions that are more stable and less sensitive to stochastic variation across individual branches.

**An Example.** An illustrative example of `DTS-Greedy` is shown in Figure 4. Given the input prompt: 'What's the area of a rectangle with length 12 and width 9?' From step 1 to $t_1 - 1$, the next-token distribution satisfies $\delta_{\text{dec}}(P_t) = 0$, thus `DTS` samples a single token per step to produce the prefix 'The area is length× width.' (blue). At step $t_1$, after feeding the token 'Here,' into the model, the next-token distribution meets $\delta_{\text{dec}}(P_{t_1}) = 1$; `DTS` therefore generates new branches (green) by selecting the top two tokens 'the' and 'length', where each branch starts with these two tokens. From step $t_1 + 1$ to $t_2 - 1$, `DTS` proceeds each branch (blue and green) with single-token sampling due to $\delta_{\text{dec}}(P_t) = 0$. At step $t_2$, after feeding the token 'So' from the green branch, the condition $\delta_{\text{dec}}(P_{t_2}) = 1$ holds again, and `DTS` expands the green branch by selecting the top two tokens 'length×' and 'area=', yielding three branches in total (blue, green, and purple). After step $t_2$, decoding continues with single-token sampling along all active branches and stops as soon as any branch emits the end token `EOS` (purple branch). The final output is 'The area is length× width. Here, length =12 and width = 9. So area= 12× 9= 108'.

### 3.3. The Algorithm of `DTS`

`DTS` is described in Algorithm 1. The algorithm begins by initializing the reasoning set and the completed trajectories set with $\varnothing$ (line 1). During the decoding, `DTS` follows Equation (3) to expands new branches (line 3); and then follows Equation (4) to update the reasoning process (line 4). At each step, any trajectories that have reached the ending token are added to the completed set (line 5). Decoding continues until the size of the completed trajectories set has reached the voting budget, at which point a final voted answer is returned (lines 6-8). For `DTS-Greedy`, the voting budget is one. Overall, `DTS` follows a breadth-first search strategy over the sketched decoding tree, ensuring that short and completed reasoning trajectories are prioritized under both `DTS-Greedy` and `DTS-Stable`.

---

**Algorithm 1** Decoding Tree Sketching (`DTS`)

**Input:** LRM $f$, input prompt $x$, voting budget $B$.
**Output:** Final reasoning answer $y^*$.

1: $\mathcal{T}_0, \mathcal{C} = \varnothing, \varnothing$
2: **for** $t = 1, 2, \cdots$ **do**
3:     Generate new branches by Eq. (3)
4:     Collect trajectory candidates $\mathcal{T}_t$ by Eq. (4)
5:     $\mathcal{C} = \mathcal{C} \cup \{\xi_t^i \in \mathcal{T}_t \mid \texttt{EOS} \in \xi_t^i\}$
6:     **if** $|\mathcal{C}| \geq B$ **then**
7:         $y^* \leftarrow$ Majority voting of candidate answers in $\mathcal{C}$
8:         **return** $y^*$
9:     **end if**
10: **end for**

---

## 4. Experiments

In this section, we conduct experiments to evaluate the performance of `DTS` framework, aiming to answer the following research questions: **RQ1**: Does `DTS` produce more

*Table 1.* Accuracy (%)↑ of DTS compared to baselines on the AIME24, AIME25, GPQA-D, and LiveBench datasets.

| Model | Method | AIME24 | | AIME25 | | GPQA-D | | LiveBench | | Average | |
|---|---|---|---|---|---|---|---|---|---|---|---|
| | | ACC | Improve | ACC | Improve | ACC | Improve | ACC | Improve | ACC | Improve |
| DS-Distill-Qwen-1.5B | Standard inference | 26.67 | 0.00 | 24.67 | 0.00 | 32.02 | 0.00 | 6.00 | 0.00 | 22.34 | 0.00 |
| | Self-Consistency | 40.67 | +14.00 | 33.33 | +8.66 | 37.58 | +5.56 | 8.00 | +2.00 | 29.89 | +7.55 |
| | DeepConf-low | 46.00 | +19.33 | 34.00 | +9.33 | 38.89 | +6.87 | 9.30 | +3.30 | 32.05 | +9.71 |
| | DeepConf-high | 44.00 | +17.33 | 36.00 | +11.33 | 36.87 | +4.85 | 12.00 | +6.00 | 32.22 | +9.88 |
| | DTS-Greedy | 54.67 | +28.00 | 34.67 | +10.00 | 41.41 | +9.39 | 17.30 | +11.30 | 37.01 | +14.67 |
| | DTS-Stable | 64.67 | +38.00 | 39.33 | +14.66 | 41.11 | +9.09 | 16.70 | +10.70 | 40.45 | +18.11 |
| DS-Distill-Qwen-7B | Standard inference | 52.67 | 0.00 | 36.00 | 0.00 | 49.29 | 0.00 | 27.20 | 0.00 | 41.29 | 0.00 |
| | Self-Consistency | 69.33 | +16.66 | 54.00 | +18.00 | 53.03 | +3.74 | 36.70 | +9.50 | 53.27 | +11.97 |
| | DeepConf-low | 70.67 | +18.00 | 52.67 | +16.67 | 52.32 | +3.03 | 33.30 | +6.10 | 52.24 | +10.95 |
| | DeepConf-high | 63.33 | +10.66 | 49.33 | +13.33 | 52.32 | +3.03 | 28.70 | +1.50 | 48.42 | +7.13 |
| | DTS-Greedy | 73.33 | +20.66 | 53.33 | +17.33 | 55.76 | +6.47 | 32.70 | +5.50 | 53.78 | +12.49 |
| | DTS-Stable | 78.67 | +26.00 | 56.67 | +20.67 | 57.78 | +8.49 | 40.70 | +13.50 | 58.45 | +17.16 |
| Qwen3-0.6B | Standard inference | 11.33 | 0.00 | 14.00 | 0.00 | 24.75 | 0.00 | 24.70 | 0.00 | 18.70 | 0.00 |
| | Self-Consistency | 11.33 | +0.00 | 25.33 | +11.33 | 25.05 | +0.30 | 27.30 | +2.60 | 22.25 | +3.56 |
| | DeepConf-low | 13.33 | +2.00 | 24.00 | +10.00 | 25.96 | +1.21 | 32.00 | +7.30 | 23.82 | +5.13 |
| | DeepConf-high | 14.67 | +3.34 | 20.67 | +6.67 | 25.56 | +0.81 | 22.70 | -2.00 | 20.90 | +2.21 |
| | DTS-Greedy | 14.00 | +2.67 | 25.33 | +11.33 | 25.56 | +0.81 | 28.60 | +3.90 | 23.37 | +4.68 |
| | DTS-Stable | 18.67 | +7.34 | 29.33 | +15.33 | 26.57 | +1.82 | 34.70 | +10.00 | 27.32 | +8.62 |
| Phi4-mini-reasoning-4B | Standard inference | 49.33 | 0.00 | 36.00 | 0.00 | 50.20 | 0.00 | 42.40 | 0.00 | 44.48 | 0.00 |
| | Self-Consistency | 61.33 | +12.00 | 42.00 | +6.00 | 52.32 | +2.12 | 55.30 | +12.90 | 52.74 | +8.26 |
| | DeepConf-low | 62.67 | +13.34 | 38.00 | +2.00 | 52.83 | +2.63 | 51.30 | +8.90 | 51.20 | +6.72 |
| | DeepConf-high | 50.00 | +0.67 | 35.33 | -0.67 | 51.72 | +1.52 | 38.70 | -3.70 | 43.94 | -0.55 |
| | DTS-Greedy | 68.00 | +18.67 | 44.00 | +8.00 | 53.13 | +2.93 | 52.80 | +10.40 | 54.48 | +10.00 |
| | DTS-Stable | 71.33 | +22.00 | 52.00 | +16.00 | 53.03 | +2.83 | 56.00 | +13.60 | 58.09 | +13.61 |

accurate reasoning? **RQ2**: Can DTS mitigate the repetitive generation during reasoning? **RQ3**: Does DTS select more reasonable trajectories?

### 4.1. Experimental Setup

We specify the models, datasets, and baseline methods below. We provide more information on the experiment implementation details in Appendix A.

**Models.** We evaluate DTS with four representative LRMs: DeepSeek-R1-Distill-Qwen-7B and -1.5B (DeepSeek-AI, 2025), Qwen3-0.6B (Team, 2025), and Phi4-mini-reasoning-4B (Abdin et al., 2024), whose pre-trained weights are from Huggingface Transformers (Wolf et al., 2020).

**Datasets.** The evaluation of DTS is based on four reasoning dataset: AIME24, AIME25 (math ai, 2025), GPQA-Diamond (Rein et al., 2023), and LiveBench-Reasoning (White et al., 2025). All datasets are accessed through the Huggingface Datasets (Lhoest et al., 2021) library. AIME24 and AIME25 consist of high-difficulty problems from the American Invitational Mathematics Examination, each containing 30 questions (Wang et al., 2024). GPQA-Diamond contains 198 graduate-level STEM questions. LiveBench-Reasoning contains 200 questions on logical deduction, puzzle solving, and spatial reasoning. We follow existing works (Chen et al., 2025; Xu et al., 2025) to construct prompts and extract answers for all datasets.

**Baseline Methods.** **Standard Inference:** We evaluate each LRM under its default inference setting. The model generates a single reasoning trace for each problem, and the final predicted answer is compared against the ground-truth label from the dataset. **Self-Consistency:** Self-Consistency (Wang et al., 2023b) allows the LRM to generate multiple reasoning trajectories independently for each problem. Each trajectory produces a candidate answer, and the final prediction is obtained by majority voting over these answers. **DeepConf:** DeepConf (Fu et al., 2025) improves multi-trajectory reasoning by using confidence scores from the model's next-token distributions to filter out low-quality traces during generation. The method first estimates a confidence threshold and continues generation while discarding traces whose confidence falls below this threshold. We evaluate DeepConf-low, which applies a stricter threshold and keeps only highly confident traces, and DeepConf-high, which uses a looser threshold to encourage exploration.

### 4.2. Performance on Reasoning Tasks (RQ1)

Table 1 shows the accuracy(%) of DTS, compared with baseline methods and standard inference of LRMs.

**Accuracy Improvement.** As shown in Table 1, DTS consistently outperforms all baseline methods and standard inference. Furthermore, the single-trajectory variant DTS-Greedy surpasses multi-trajectory baselines such as Self-Consistency and DeepConf in most cases, indicating that selectively branching at decision tokens is more effective than generating more independent samples. These results show that by sketching the decoding tree and with early termination, DTS steers decoding toward existing optimal solutions in the reasoning space.

*Table 2.* Repetition rate (%) ↓ of Standard Inference and `DTS-Greedy` on different models and datasets.

| Model | Dataset | Std. Inf. | DTS-Greedy |
|---|---|---|---|
| DS-Distill-Qwen-1.5B | AIME24 | 15.33 | **4.67** |
| | AIME25 | 26.67 | **6.00** |
| | GPQA-D | 1.40 | **0.00** |
| | LiveBench | 26.40 | **0.00** |
| | Average | 17.45 | **2.67** |
| DS-Distill-Qwen-7B | AIME24 | 6.67 | **1.33** |
| | AIME25 | 12.67 | **2.67** |
| | GPQA-D | 0.50 | **0.00** |
| | LiveBench | 17.20 | **0.00** |
| | Average | 9.26 | **1.00** |
| Qwen3-0.6B | AIME24 | 5.33 | **0.00** |
| | AIME25 | 4.00 | **0.00** |
| | GPQA-D | 1.00 | **0.00** |
| | LiveBench | 2.00 | **0.00** |
| | Average | 3.08 | **0.00** |
| Phi4-mini-reasoning-4B | AIME24 | 6.00 | **0.00** |
| | AIME25 | 9.33 | **0.00** |
| | GPQA-D | 1.20 | **0.00** |
| | LiveBench | 2.80 | **0.00** |
| | Average | 4.83 | **0.00** |

*Table 3.* Accuracy (%) ↑ comparison of confidence-based log-probability trajectory selection and `DTS`'s early stopping selection on different models and datasets.

| Dataset | Model | Max Logits | DTS-Greedy |
|---|---|---|---|
| AIME24 | DS-Distill-Qwen-1.5B | 38.67 | **54.67** |
| | DS-Distill-Qwen-7B | 65.33 | **73.33** |
| | Qwen3-0.6B | 7.33 | **14.00** |
| | Phi4-mini-reasoning-4B | 58.00 | **68.00** |
| AIME25 | DS-Distill-Qwen-1.5B | 21.33 | **34.67** |
| | DS-Distill-Qwen-7B | 50.00 | **53.33** |
| | Qwen3-0.6B | 16.67 | **25.33** |
| | Phi4-mini-reasoning-4B | 42.00 | **44.00** |

**Model Agnosticism.** By integrating with different families of LRMs, `DTS` delivers consistent performance, as shown in Table 1. Since it operates purely at inference time and requires no model post-training, its improvements transfer across different model architectures. This indicates that `DTS` can potentially serve as a versatile enhancement for a range of LRMs in practice.

**Task Generalization.** Beyond math, `DTS` demonstrates strong performance across diverse reasoning domains. `DTS` exhibits strong and stable accuracy gains in both GPQA-D and LiveBench in Table 1. The domains from these datasets span across physics, chemistry, biology, semantic understanding, and logical reasoning. These robust gains indicate `DTS` provides a general enhancement across diverse subjects and tasks.

### 4.3. Mitigation of Repetitive Reasoning (RQ2)

In this section, we show that `DTS` reduces repetition in LRM decoding. The repetition problem is a special case of

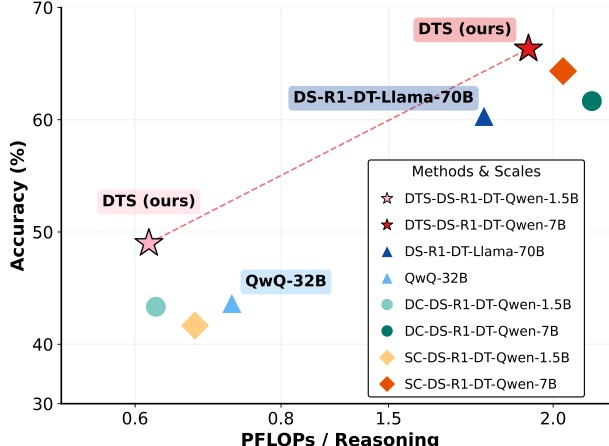

*Figure 5.* Average Accuracy (%) ↑ vs. PFLOPs ↓ of `DTS` and baseline methods compared with flagship LRMs DS-Distill-Llama-70B and QwQ-32B-Preview on the AIME24 and AIME25 datasets.

overthinking where the model falls into a reasoning loop and continuously generates repeating phrases or tokens without concluding. This phenomenon signals degraded reasoning quality, as the model stagnates on intermediate steps and accumulates inconsistencies (Yao et al., 2025; Pipis et al., 2025). Additionally, as recent studies (Xie et al., 2025) indicate, once repetition occurs, LRMs are unlikely to untrap themselves and exploit all decoding token budget with no final answer. `DTS` provides a solution to this failure mode intrinsically by sketching the reasoning tree and favoring the shorter completed trajectories. Trajectories that fall into repetition are overridden by concise completions, preventing the decoder from returning repeated segments.

Table 2 reports the rate of cases where repetition occurred under standard inference and `DTS-Greedy`, where the repetition is identified by the LRM reaching the maximum token budget. Across all datasets and models, `DTS-Greedy` consistently reduces the repetition rate, demonstrating its ability to recover from endless generation. These results confirm `DTS` sketches toward more reliable and consistent optimal solutions.

### 4.4. DTS versus Max-logit Trajectory Selection

We compare our shortest trajectory selection with confidence-based selection for the final answer in this section. Specifically, a common practice for confidence-based selection is to use LRM's cumulative logit as the decision indicator (Leang et al., 2025b; Just et al., 2025; Xu et al., 2025), selecting the trajectory that the model assigns the highest confidence. However, this assumption has been widely challenged in recent studies (Simhi et al., 2025; Kumar et al., 2024; Steyvers et al., 2025; Wang et al., 2023a), showing that models can assign high probability to fluent yet logically incorrect traces, and that confidence can increase with repetition or local consistency rather than global correctness. In contrast, our shortest trajectory selection

**Question**: There exist real numbers $x$ and $y$, both greater than 1, such that $\log_x(y^x) = \log_y(x^{4y}) = 10$. Find $xy$.

**DTS Response**: Alright, so I have this problem here: there exist real numbers x and y, both greater than 1, such that log base $x$ of $y^x$ equals 10, and log base y of $x^{4y}$ also equals 10 ... $x^{10} = y^x$ ... $y^{10} = x^{4y}$ ... $\ln(x^{10}) = \ln(y^x)$ ... $10\ln x = x\ln y$ ... $\ln(y^{10}) = \ln(x^{4y})$ ... $10\ln y = 4y\ln x$ ... We can divide both sides by $\ln x : \left[\frac{10}{x} = \frac{2y}{5}\right]$. Solving for $y$: $[10 \cdot 5 = 2xy \Rightarrow 50 = 2xy \Rightarrow xy = 25]$. Thus, the product $xy$ is $\boxed{25}$. 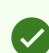

**Standard Inference Response**: Okay, so I have this problem here: ... $x^{10} = y^x$ ... $y^{1}0 = x^{4y}$ ... Taking ... $10\ln x = x\ln y$ ... $10\ln y = 4y\ln x$ ... $25/e^a = e^b$ ... $y = 25/x$ ... So, let me plug into the original equations. So, $10\ln x = x(\ln 25 - \ln x)$ ... So, $10\ln x = 25\ln x - x\ln x$ ... $x = 15$ ... check ... $\neq 10$ ... Wait, perhaps I made a mistake... Alternatively, perhaps the problem is to find $x$ and $y$ such that $\log_x(y^x) = 10$ and $\log_y(x^y) = 5$, ... Thus, $y = x^{10/x}$, $x = y^{5/x}$ ... Thus, $x = \left(x^{10/x}\right)^{5/x} = x^{50/x^2}$ ... Thus, $x = 5\sqrt{2}$ ... check ... $\neq 5$ ... Thus, discrepancy… Thus, $x = 5\sqrt{2}$ ... check ... $\neq 5$ ... Thus, discrepancy… Thus, the answer is $x = 5$, $y = 5$, but they don't satisfy. Alternatively, perhaps the problem is to find $x$ and $y$ such that $\log_x(y^x) = 10$ and $\log_y(x^y) = 5$… Thus, the answer is $x$ and $y$, but they don't satisfy…
**No conclusion reached due to repetition.** 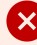

*Figure 6.* A case study illustrating how `DTS` produces a correct solution, while standard inference enters an endless repetition loop, failing to reach a conclusion after consuming all token budgets.

strategy is motivated by both the theoretical aspects of the LRM's post-training scheme and the empirically observed anti-correlation between reasoning length and accuracy in Section 2.2. Table 3 presents the performance comparison between confidence-based max-logit selection and shortest-trajectory selection across all models and datasets[2]. We observe that our selection strategy consistently achieves higher accuracy in every setting. These results support that selecting the shortest completed trace is a better criterion than confidence-based scoring.

### 4.5. Improving Inference-time Scaling

`DTS` achieves better inference-time scaling by delivering higher performance per unit of compute. We report the average accuracy and computational footprint for all models and baselines on AIME24 and AIME25 in Figure 5.

**Comparison with Flagship LRMs.** We first compare smaller LRMs utilizing `DTS-Stable` with industry-scale flagship LRMs under matched compute budgets. At similar PFLOPs levels, DS-Distill-Qwen-1.5B and 7B achieve higher accuracy than QwQ-32B-Preview and DS-Distill-Llama-70B, respectively. This demonstrates that `DTS` enhances smaller LRMs to reach flagship-level reasoning performance under identical inference-scaling conditions.

**Comparison with Parallel Thinking Baselines.** We further compare `DTS` with existing parallel thinking methods applied to the same LRMs. `DTS` consistently achieves higher accuracy at lower PFLOPs than Self-Consistency and DeepConf. These results highlight the effectiveness of `DTS`'s structured tree sketching and reliable trajectory selection, yielding a more favorable accuracy-compute scaling than baseline methods.

---
[2]We follow the Beam Search method (Wu et al., 2016) to support max-logit selection with length normalization.

*Table 4.* Accuracy (%) ↑ of `DTS-Stable` under different threshold value combinations.

| Model | Dataset | $\tau_v=1.5, \tau_h=1.2$ | $\tau_v=1.5, \tau_h=2.5$ | $\tau_v=1, \tau_h=2.5$ |
|---|---|---|---|---|
| DS-Distill Qwen-1.5B | AIME24 | 64.00 | **64.67** | 64.67 |
| | AIME25 | 38.67 | **39.33** | 39.33 |
| DS-Distill Qwen-7B | AIME24 | 76.67 | **78.67** | 78.00 |
| | AIME25 | **57.33** | 56.67 | 55.33 |

### 4.6. Demonstration of `DTS` Reasoning (RQ3)

To further illustrate how `DTS` enhances reasoning behavior, we present a case study in Figure 6 from the AIME25 dataset using the DS-Distill-Qwen-1.5B model. Under standard inference, the model repeatedly revisits the same partial calculations ultimately failing to reach a conclusion. This behavior corresponds to the repetitive reasoning phenomenon discussed in Section 4.3, where the repeating segments are highlighted in red. In contrast, `DTS` sketches the decoding tree around decision tokens and settles on a coherent reasoning trajectory that leads to the correct answer. The final trace remains concise, logically structured, and aligned with the correct solution, illustrating how `DTS` helps the model avoid reasoning loops and converge on a reliable trajectory. We provide more case studies in Appendix B.

### 4.7. Sensitivity to Threshold Parameters

To evaluate the robustness of `DTS` to the threshold parameters, we analyze the effect of varying the varentropy threshold $\tau_v$ and entropy threshold $\tau_h$ on `DTS`'s performance. As shown in Table 4, `DTS` with $\tau_v = 1.5, \tau_h = 2.5$ achieves better performance across different settings, indicating its effectiveness. Furthermore, performance remains stable across nearby threshold values, demonstrating that `DTS` is robust and relatively insensitive to parameter selection. We provide additional sensitivity and decision token criteria analysis in Appendix C.

## 5. Related Works

**Parallel Thinking.** Parallel thinking methods, such as Self-Consistency (Wang et al., 2023b), Best-of-N (Brown et al., 2024), EGB (Li et al., 2026), and DeepConf (Fu et al., 2025), enhance LRMs by generating multiple reasoning trajectories and aggregating their solutions through a voting or scoring mechanism. By exploring various reasoning, these methods reduce the impact of individual reasoning errors and improve overall inference robustness. As a result, parallel thinking has become a widely adopted inference-time technique for improving LRM reasoning performance.

**Adaptive Thinking.** Another line of research studies supervised fine-tuning (SFT) and RL methods that adapt reasoning length during the training phase. For instance, AutoL2S (Luo et al., 2025), DAST (Shen et al., 2025), and L1 (Aggarwal & Welleck, 2025) optimize models to generate either short or long responses based on the perceived difficulty of the prompt. However, these adaptive thinking methods incur additional training and data curation costs and are unable to improve LRM reasoning accuracy (Wang et al., 2025).

## 6. Conclusion

In this work, we introduced Decoding Tree Sketching (`DTS`), a training-free decoding framework for enhancing LRM reasoning. `DTS` enables structured reasoning explorations by sketching a compact decoding tree and leveraging the length-accuracy anti-correlation to select short yet reliable reasoning for the final solution. `DTS` branches selectively at decision tokens to maintain trajectory diversity. Motivated by a proven length-accuracy anti-correlation, `DTS` prioritizes high-quality solutions via early termination. Experiments across four models and four datasets comprehensively show stable accuracy gains and mitigation of repetition. These results show that `DTS` enhances LRM reasoning through structured exploration and reasoning selection.

## Acknowledgements

This research was supported by NSF CNS 2528780. We would like to thank The Center for Language and Speech Processing at Johns Hopkins University for providing hardware and technical assistance for experiments. The views and conclusions in this paper are those of the authors and do not represent the views of any funding or supporting agencies.

## Impact Statement

This paper presents work whose goal is to advance the field of Machine Learning. There are many potential societal consequences of our work, none which we feel must be specifically highlighted here.

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

## A. Implementation Details

For `DTS`, we set varentropy threshold $\tau_v = 1.5$, entropy threshold $\tau_h = 2.5$, and $K = 3$ across all datasets and models. To prevent the decoding tree from growing exponentially, we additionally limit the number of simultaneously active branches at 48. Once this limit is met, `DTS` temporarily disables branching and always applies the second case of Equation 3, even if $\delta_{\text{dec}}(P_t) = 1$. This design prioritizes earlier decision tokens that determine the high-level shape of the sketched decoding tree, while suppressing later ones that mostly dive into minor variations of already-established branches. For multi-trajectory methods with majority voting, including Self-Consistency, DeepConf, and `DTS-Stable`, we set the voting budget $N = 8$. For all experiments, we use the LRM's recommended hyperparameter settings and maximum token budget. Stochasticity is controlled by fixing random seeds $s \in \{0, 1, 2, 3, 4\}$ and the average accuracy over these five runs is reported.

## B. More Case Studies

Figure 7, 8, 9, 10, and 11 present additional qualitative case studies spanning all evaluated models and datasets, including AIME24, AIME25, GPQA-D, and LiveBench. Across these diverse reasoning tasks, a consistent pattern emerges: under standard inference, models frequently fall into repetitive or self-contradictory reasoning loops, revisiting partial computations or local deductions without converging to a final answer. In contrast, `DTS` produces reasoning trajectories that are shorter, better structured, and more aligned with correct solutions. These examples further reinforce the robustness of `DTS`, complementing the quantitative improvements reported in Table 1.

**Question**: The 9 members of a baseball team went to an ice-cream parlor after their game. Each player had a single scoop cone of chocolate, vanilla, or strawberry ice cream. At least one player chose each flavor, and the number of players who chose chocolate was greater than the number of players who chose vanilla, which was greater than the number of players who chose strawberry. Let *N* be the number of different assignments of flavors to players that meet these conditions. Find the remainder when *N* is divided 1000.

**DTS Response**: Alright, so I've got this problem...Let me denote the number of players choosing chocolate as c, vanilla as v, and strawberry as s... So, I need to find all triples (c, v, s) such that c > v > s, each at least 1, and c + v + s = 9.... Therefore, the problem reduces to finding all triples (c, v, s)... and then summing 9!/(c! v! s!) for each triple.... So, overall, the valid triples are: (6,2,1), (5,3,1), (4,3,2).... For (6, 2, 1): ... 252.... For (5, 3, 1): ... 504.... For (4, 3, 2): ... 1260.... Summing these values gives the total number of assignments: ... 2016.... Finally, we find the remainder when 2016 is divided by 1000: ... 16.... Thus, the answer is $\boxed{16}$. ✅

**Standard Inference Response**: Okay, so I have this problem... So, the number of such assignments is ... the sum over all such triples of 9! / (C! V! S!)...Alternatively, perhaps it's easier to list all possible triples (C, V, S) where C > V > S ≥ 1 and C + V + S = 9... Thus, total of three solutions: (6,2,1), (5,3,1), and (4,3,2)... Let me define variables x, y, z such that x = C - 2, y = V - 1, z = S...But S = 0 is invalid ...So, only one solution...Wait, but earlier, when I tried to compute via substitution, I found only one solution, but now, by enumerating, I find three... So, perhaps substitution is missing some solutions.... Wait, according to the formula, the number of partitions of 9 into 3 distinct parts is 5. ... Thus, N=5.... But according to substitution, it's 3. ... Thus, substitution is wrong... Thus, the correct number is 5.... But according to the formula, it's 5. ... Thus, N=5. ... Thus, the answer is 5.... And according to substitution, it's 3. ... Thus, substitution is wrong. ... Thus, the correct number is 5.... Thus, N=5. ... Thus, N mod1000=5. ... Thus, the answer is 5.... But according to substitution, it's 3. ... Thus, substitution is wrong. ... Thus, the correct number is 5.... Thus, N=5. ... Thus, N mod1000=5. ... Thus, the answer is 5.... But according to substitution, it's 3. ... Thus, substitution is wrong. ... Thus, the correct number is 5....But according to substitution…
**No conclusion reached due to repetition.** ❌

*Figure 7.* A case study from AIME25 dataset using DS-Distill-Qwen-1.5B with `DTS` and standard inference.

**Question**: Let *N* be the greatest four-digit positive integer with the property that whenever one of its digits is changed to 1, the resulting number is divisible by 7. Let *Q* and *R* be the quotient and remainder, respectively, when *N* is divided by 1000. Find *Q+R*

**DTS Response**: To find the greatest four-digit number N such that changing any of its digits to 1 results in a number divisible by 7, we set up congruences for N = 1000A + 100B + 10C + D. The four conditions translate to 1000 + 100B + 10C + D ≡ 0 mod (7) ... and 1000A + 100B + 10C + 1 ≡ 0 mod (7). Converting these modulo 7 yields a system including 6 + 2B + 3C + D ≡ 0 mod (7) ... Solving these congruences reveals A ≡ 5 mod (7) ... Substituting A = 5 gives B = 6, C = 9, and D = 4. Thus, N = 5694. When N is divided by 1000, the quotient Q is 5 and the remainder R is 694. Therefore, Q + R = 5 + 694 = 699.\n\n $\boxed{609}$ ✅

**Standard Inference Response**: Okay, let's try to tackle this problem step by step... The question is about finding the greatest four-digit number N such that if you change any one of its digits to 1, the resulting number is divisible by 7... we can express d2, d3, d4 in terms of d1... Start with d1=9... Compute d2≡4 mod7... So d2=4... d3≡3 mod7... So d3=3... d4≡0 mod7... So N=9437... But wait, let's verify if this N=9437 satisfies all four original conditions... Change thousands digit to 1: 1437... 1437 ÷7=205.285... 7205=1435, 1437-1435=2. Not divisible by 7. So this fails... Therefore, the equations (1)-(4) must hold, but N itself doesn't need to be 0 mod7... But in our case... we get N=9437. Change thousands digit to 1:1437…Change thousands digit to 1:1996…Change thousands digit to 1:1926…Change thousands digit to 1:1924…Change thousands digit to 1:1694…Change thousands digit to 1:1624…Not divisible.\n\nAlmost there, but one failure.\n\nNext, a=5, b=6, c=2, d=4: N=5624. Checked.\n\nNext, a=5, b=6, c=2, d=4: N=5624. Checked.\n\nNext, a=5, b=6, c=9, d=4: N=5694. Checked.\n\nNext, a=5, b=6, c=2, d=4: N=5624. Checked.\n\nNext, a=5, b=6, c=9, d=4: N=5694… **No conclusion reached due to repetition.** ❌

*Figure 8.* A case study from AIME24 dataset using Phi4-mini-reasoning-4B with `DTS` and standard inference.

**Question**: In this question, assume each person either always tells the truth or always lies. The person at the campground thinks their friend is lying. Mateo is at the aquarium. The person at the restaurant says the person at the hotel lies. Farid is at the movie theater. The person at the movie theater says the person at the campground lies. Ryan is at the shopping mall. The person at the cafe says the person at the campground lies. The person at the observatory says the person at the museum lies. The person at the museum says the person at the restaurant tells the truth. The person at the campground says the person at the airport tells the truth. The person at the campground tells the truth. Luna is at the restaurant. The person at the movie theater saw a firetruck. Isabella is at the observatory. The person at the amusement park says the person at the museum tells the truth. The person at the restaurant says the person at the shopping mall tells the truth. Liam is at the beach. Ayaan is at the airport. The person at the restaurant says the person at the beach tells the truth. Maya is at the amusement park. The person at the restaurant says the person at the aquarium lies. The person at the restaurant says the person at the movie theater lies. The person at the library says the person at the campground tells the truth. Elowen is at the hotel. Priya is at the library. The person at the gym says the person at the restaurant tells the truth. Kehinde is at the cafe. Anika is at the museum. Quan is at the campground. Soren is at the gym. Does the person at the museum tell the truth? Does the person at the restaurant tell the truth? Does the person at the movie theater tell the truth? Think step by step, and then put your answer in **bold** as a list of three words, yes or no (for example, **yes, no, yes**). If you don't know, guess.

**DTS Response**: Alright, let's try to figure this out step by step. So, we have a bunch of people at different places, each either always telling the truth or always lying. We need to figure out if the person at the museum, the restaurant, and the movie theater tell the truth. ... Now, the person at the restaurant (R) says the person at the hotel (H) lies. The person at the hotel is the one at the observatory (O). So, the R person says H (O) lies, meaning O lies. But the person at the observatory (O) says M (museum) lies. So, that means M tells the truth. So, M tells the truth. ... Wait, let me recap: ... Person at observatory (O) says M lies, and since O lies, M tells the truth. ... Person at movie theater (M) says C lies. Since C tells the truth, M lies. ... Person at restaurant (R) says H (O) lies. Since O lies, R tells the truth. Wait, but the question is about the museum, restaurant, and movie theater. So, based on what we have: ... Museum (M): M tells the truth. ... Restaurant (R): R tells the truth. ... Movie Theater (M): M lies. So, the answers are yes, yes, no. ... So, the list is yes, yes, no. \n</think>\n\n**yes, yes, no** ✅

**Standard Inference Response**: Alright, let's try to figure out who is telling the truth. ... Beach: - Person H says: The person at the restaurant tells the truth. ... Therefore, the person at the beach tells the truth, and the person at the restaurant lies. ... The person at the restaurant says the person at the campground lies. If the restaurant person is lying, then the person at the campground tells the truth. The person at the campground says the person at the airport tells the truth and the person at the restaurant lies. ... Since the restaurant person lies, the person at the airport is lying. ... The person at the shopping mall tells the truth. ... The cafe person tells the truth. ... Now, the person at the beach is Quan, who tells the truth.\n\nThe person at the beach says the person at the restaurant tells the truth, which is false, …The person at the beach says the person at the restaurant tells the truth, which is false, …Now, the person at the beach is Quan, who tells the truth.\n\nThe person at the beach says the person at the restaurant tells the truth, which is false, …The person at the beach says the person at the restaurant tells the truth, which is false, … **No conclusion reached due to repetition.** ❌

*Figure 9.* A case study from LiveBench dataset using DS-Distill-Qwen-1.5B with `DTS` and standard inference.

**Question**: You have an interesting drought-resistant cultivar of barley, which, unfortunately, contains an anti-nutritional compound. This compound makes it impossible to use this cultivar for food purposes. After an extensive investigation, you discover the gene responsible for the synthesis of this compound. This gene consists of five exons and four introns. You decide to use old-school approaches and produce a collection of mutants using EMS chemical mutagen. You sequence a target gene in all mutant lines and discover some changes at the beginning of its sequence in the first exon area. Which of the following mutations will most probably let you eliminate the anti-nutritional compound in the cultivar?\n\nIntactgene:\n5'-ATGTTTCTCGCTGGTACTTCTGTGGATGAACATATTTATTGTCGT…TGA-3'\n\nMutant1:\n5'-ATGTTCTACGCTGGTACTTCTGTGGATGAACATATTTATTGTCGC…TGA-3'\nMutant2:\n5'-ATGTTCTAAGCTGGTACTTCTGTGGATGAACATATTTATTGTCGC…TGA-3'\nMutant3:\n5'-ATGTTTTACGCTGGTGTCACTTCTGTGGATGAACATATTTATTGTCGT…TGA-3'\nMutant4:\n5'-ATGTTTTACGCTACTTCTGTGGATGAACATATTTATTGTCGT…TGA-3'

**DTS Response**: Okay, so I've got this genetics question here about mutations in a gene and whether they can eliminate an anti-nutritional compound. ... the anti-nutrient is likely encoded by this gene ... so if the gene is disrupted ... the compound isn't produced. ... Comparing the original and each mutant ... which mutation causes a frameshift or a premature stop codon ... Mutant 1 ... a single nucleotide change ... missense ... reading frame is maintained ... Mutant 3 ... missense ... Mutant 4 ... missense ... Only Mutant2 ... the third codon is TAA (stop codon) ... a premature stop after three codons ... the protein ends ... non-functional ... so the plant would not have the compound ... the correct answer would be D) Mutant2 ✅

**Standard Inference Response**: Okay, so I've got this genetics question here about mutations and how they might affect the production of an anti-nutritional compound in barley. ... The goal is to determine which mutation would disrupt the gene function ... so if the gene is nonfunctional, the compound won't be made. ...I need to see if the mutation causes a frameshift or a premature stop ... Stop codons are UAA, UAG, UGA in mRNA ... the coding strand is same as mRNA except T→U ... so a mutation that creates a stop codon early would truncate the protein ... Mutant 1 ... lots of changes ... likely missense/reading frame maintained ... Mutant 2 ... introduces a stop codon (TAA in DNA) early ... the protein would be truncated/nonfunctional ... so the anti-nutritional compound wouldn't be produced…Let me see: the original gene's mRNA (from the coding strand, replacing T with U) would be:\n\n5'-AUGUUUUUUUAGCUAGCUAAGCUAGCUAAGCUAGCUAAGCUAGCUAAGCUAGCUAAGCUAGCUAAGCUAGCUAAGCUAGCUAAGCUAGCUAAGCUAGCUAAGCUAGCUAAGCUAGCUAAGCUAGCUAAGCUAGCUAAGCUAGCUAAGCUAGCUAAGCUAGCUAAGCUAGCUAAGCUAGCUAAGCUAGCUAAGCUAGCUAAGCUAGCUAAGCUAGCUAAGCUAGCUAAGCUAGCUAAGCUAGCUAAGCUAGCUAAGCUAGCUAAGCUAGCUAAGCUAGCUAAGCUAGCUAAGCUAGCUAAGCUAGCUAAGCUAGCUAAGCUAGCUAAGCUAGCUAAGCUAGC... **No conclusion reached due to repetition.** ❌

*Figure 10.* A case study from GPQA-D dataset using DS-Distill-Qwen-7B with `DTS` and standard inference.

**Question**: The product\\[\\prod^{63}_{k=4} \frac{\\log_k (5^{k^2 - 1})}{\\log_{k + 1} (5^{k^2 - 4})} = \frac{\\log_4 (5^{15})}{\\log_5 (5^{12})} \\cdot \frac{\\log_5 (5^{24})}{\\log_6 (5^{21})}\\cdot \frac{\\log_6 (5^{35})}{\\log_7 (5^{32})} \\cdots \frac{\\log_{63} (5^{3968})}{\\log_{64} (5^{3965})}\\] is equal to $tfrac\ mn$ where $m$ and $n$ are relatively prime positive integers. Find $m + n$

**DTS Response**: Okay, so I need to solve this product expression:\n\n$$\n\prod^{63}{k=4} \frac{\log_k (5^{k^2 - 1})}{\log{k + 1} (5^{k^2 - 4})}\n$$\n\n... Using the power rule, $\log_k (5^{k^2 - 1})=(k^2-1)\log_k 5$ and $\log_{k+1}(5^{k^2-4})=(k^2-4)\log_{k+1}5$, so each term is $\frac{k^2-1}{k^2-4}\cdot\frac{\log_k 5}{\log_{k+1}5}$ ... with $k^2-1=(k-1)(k+1)$ and $k^2-4=(k-2)(k+2)$, we get $\prod_{k=4}^{63}\frac{(k-1)(k+1)}{(k-2)(k+2)}=\left(\prod_{k=4}^{63}\frac{k-1}{k-2}\right)\left(\prod_{k=4}^{63}\frac{k+1}{k+2}\right)=(3/2)(4/3)\cdots(62/61)\cdot(5/6)(6/7)\cdots(64/65)=31\cdot\frac{1}{13}=\frac{31}{13}$ ... and $\frac{\log_k 5}{\log_{k+1}5}=\frac{\log(k+1)}{\log k}$ so $\prod_{k=4}^{63}\frac{\log(k+1)}{\log k}=\frac{\log 64}{\log 4}=3$ ... Therefore, the entire original product is $\frac{31}{13}\cdot 3=\frac{93}{13}$ ... so $m+n=93+13=106$ ... Final Answer $\boxed{106}$ ✅

**Standard Inference Response**: Note the product is from $k = 4$ to $k = 63$... Each term can be rewritten using logarithm power rules:$$\frac{(k^2 - 1) \cdot \log_k 5}{(k^2 - 4) \cdot \log_{k + 1} 5}$$...Separate the terms into factors...$$\prod_{k=4}^{63} \frac{k^2 - 1}{k^2 - 4} \cdot \frac{\log_k 5}{\log_{k + 1} 5}$$...First, handle the rational product...$$\frac{\log_k 5}{\log_{k + 1} 5} = \frac{\log 5 / \log k}{\log 5 / \log (k + 1)} = \frac{\log (k + 1)}{\log k}$$...This is a telescoping product:$$\frac{\log 5}{\log 4} \times \frac{\log 6}{\log 5} \times \dots \times \frac{\log 64}{\log 63} = \frac{\log 64}{\log 4}$$...Since $64 = 4^3$, this simplifies to $3$...Combining the parts...$$\left( \prod_{k=4}^{63} \frac{k^2 - 1}{k^2 - 4} \right) \times 3$$...But the actual product is 10/7. This suggests a contradiction…This is equal to [log_5 5 / log_5 4] * [log_5 6 / log_5 5] * ... * [log_5 64 / log_5 63].\n\nThis is equal to [log_5 5 / log_5 4] * [log_5 6 / log_5 5] * ... * [log_5 64 / log_5 63].\n\nThis is equal to [log_5 5 / log_5 4] * [log_5 6 / log_5 5] * ... This is equal to [log_5 5 / log_5 4] * [log_5 6 / log_5 5] * ... * [log_5 64 / log_5 63].\n\nThis is equal to [log_5 5 / log_5 4] * [log_5 6 / log_5 5] * ... * [log_5 64 / log_5 63].\n\nThis is equal to [log_5 5 / log_5 4] * [log_5 6 / log_5 5] * ...    **No conclusion reached due to repetition.** ❌

*Figure 11.* A case study from AIME25 dataset using Qwen3-0.6B with `DTS` and standard inference.

## C. Decision Token Criteria Analysis

Table 5 compares the performance of using different decision token criteria and threshold values in Equation 3 on AIME25 with DS-Distill-Qwen-1.5B. We show that `DTS`'s high varentropy and low entropy criteria outperform other combinations.

*Table 5.* Accuracy (%) ↑ comparison of different decision token criteria and threshold values on AIME25 and DS-Distill-Qwen-1.5B.

| Criteria | Threshold | DTS-Greedy | DTS-Stable |
|---|---|---|---|
| $\mathrm{H}(P_t) \geq \tau_\mathrm{h}$ | $\tau_\mathrm{h}{=}2.5$ | 29.33 | 33.33 |
| $(\mathrm{VE}(P_t) \geq \tau_\mathrm{v}$ and $\mathrm{H}(P_t) \leq \tau_\mathrm{h})$ or $\mathrm{H}(P_t) \geq \tau_\mathrm{h}$ | $\tau_\mathrm{v}{=}1.5,\ \tau_\mathrm{h}{=}2.5$ | 30.67 | 34.67 |
| $\mathrm{VE}(P_t) \geq \tau_\mathrm{v}$ and $\mathrm{H}(P_t) \leq \tau_\mathrm{h}$ | $\tau_\mathrm{v}{=}1.5,\ \tau_\mathrm{h}{=}2.5$ | **34.67** | **39.33** |
| $\mathrm{VE}(P_t) \geq \tau_\mathrm{v}$ and $\mathrm{H}(P_t) \leq \tau_\mathrm{h}$ | $\tau_\mathrm{v}{=}1.5,\ \tau_\mathrm{h}{=}1.2$ | **34.67** | 38.67 |

## D. More on the Length-Accuracy Anti-Correlation

To further validate the length-accuracy anti-correlation observed in Section 2.2 beyond the influence of question difficulty, we report a within-question macro accuracy that debiases the effect of varying problem difficulties across questions. Specifically, macro accuracy is estimated by:

$$Acc_{\mathrm{macro}}(b) = \frac{1}{N} \sum_{i=1}^{N} \frac{\sum_{j \in b} \mathbb{I}[\mathrm{correct}_{i,j}]}{|\{j \in b\}|} \tag{5}$$

where $b$ denotes a length bin and $N$ is the number of questions. By normalizing accuracy within each question, this metric debiases the influence of question difficulty on reasoning length. Table 6 reports the macro-averaged accuracy (%) within each length bin. Across all models and datasets, accuracy consistently decreases as reasoning length grows, confirming the length-accuracy anti-correlation is not a result of question difficulty.

## E. Comparison with Additional Sampling Baselines

To further evaluate the effectiveness of `DTS`, we compare `DTS-Greedy` with top-$k$ and top-$p$ sampling (Wang & Zhou, 2024). As shown in Table 7, `DTS-Greedy` consistently outperforms both sampling methods, demonstrating that the structured tree exploration and early termination strategy of `DTS` yield more reliable reasoning trajectories than stochastic sampling methods.

*Table 6.* Macro-averaged accuracy (%) ↑ within length bins. Each value represents the within-question averaged accuracy for trajectories falling in the corresponding token length range.

| Model | Dataset | $2\times10^3$ | $4\times10^3$ | $8\times10^3$ | $12\times10^3$ |
|---|---|---|---|---|---|
| DS-Distill-Qwen-1.5B | AIME24 | **79.2** | 61.4 | 49.7 | 22.1 |
| | AIME25 | **83.3** | 54.4 | 31.1 | 20.5 |
| DS-Distill-Qwen-7B | AIME24 | **98.3** | 84.7 | 68.8 | 46.5 |
| | AIME25 | **83.3** | 68.6 | 47.2 | 31.1 |

*Table 7.* Accuracy (%) ↑ comparison of `DTS-Greedy` against top-$k$ and top-$p$ sampling baselines.

| Model | Dataset | DTS-Greedy | top-$k$ | top-$p$ |
|---|---|---|---|---|
| DS-Distill-Qwen-1.5B | AIME24 | **54.67** | 43.33 | 34.67 |
| | AIME25 | **34.67** | 29.33 | 23.33 |
| DS-Distill-Qwen-7B | AIME24 | **73.33** | 69.33 | 66.67 |
| | AIME25 | **53.33** | 43.33 | 42.67 |

# F. Ablation Study

To further understand the contribution of each component in `DTS`, we provide an ablation study in Table 8 by disabling the structured exploration module and the shortest trajectory selection module independently. Specifically, "shortest w/o expl." disables the decoding tree exploration and instead performs independent sampling, while retaining the shortest trajectory selection. "w/o shortest w/o expl." disables both modules, reducing `DTS` to standard inference. As shown in Table 8, both modules contribute positively to the performance gains of DTS-Greedy. Together, these results confirm that the two modules are complementary and both are essential to the overall effectiveness of DTS.

*Table 8.* Accuracy (%) ↑ of DTS-Greedy under ablation of the exploration and shortest trajectory selection modules.

| Model | Dataset | DTS-Greedy | shortest w/o expl. | w/o shortest w/o expl. |
|---|---|---|---|---|
| DS-Distill-Qwen-1.5B | AIME24 | **54.67** | 43.33 | 26.67 |
| | AIME25 | **34.67** | 31.33 | 24.67 |
| DS-Distill-Qwen-7B | AIME24 | **73.33** | 66.67 | 52.67 |
| | AIME25 | **53.33** | 52.67 | 36.00 |

# G. Experiments on Larger LRMs

To validate the effectiveness of `DTS` beyond the model scales reported in Table 1, we conduct additional experiments on two larger LRMs: Qwen3-14B (Team, 2025) and DS-Distill-Qwen-14B (DeepSeek-AI, 2025). As shown in Table 9, DTS consistently outperforms all baseline methods on both larger models, demonstrating that the structured exploration and early termination strategy of `DTS` generalizes effectively to larger model scales.

# H. Computational Infrastructure

The computational infrastructure information is given in Table 10.

*Table 9.* Accuracy (%) ↑ of `DTS` compared to baseline methods on larger LRMs.

| Model | Method | AIME24 | AIME25 |
|---|---|---|---|
| Qwen3-14B | Standard Inference | 76.67 | 63.33 |
| | Self-Consistency | 76.67 | 66.67 |
| | DeepConf | 80.00 | 73.33 |
| | DTS-Greedy | 80.00 | 70.00 |
| | **DTS-Stable** | **86.67** | **76.67** |
| DS-Distill-Qwen-14B | Standard Inference | 65.33 | 48.00 |
| | Self-Consistency | 83.33 | 60.00 |
| | DeepConf | 80.00 | 60.00 |
| | DTS-Greedy | 80.00 | 63.33 |
| | **DTS-Stable** | **86.67** | **70.00** |

*Table 10.* Experiment configuration and computing infrastructure.

| Name | Value |
|---|---|
| Data type | `torch.bfloat16` |
| Flash-Attention | False |
| Eval batch-size | 1 |
| Computing Infrastructure | GPU |
| GPU Model | NVIDIA-H200 |
| GPU Memory | 141GB |
| GPU Number | 4 |
| CUDA Version | 12.4 |
| CPU Memory | 512GB |

