# OpenReview forum: "DTS: Enhancing Large Reasoning Models via Decoding Tree Sketching"
_ICML.cc/2026/Conference — ICML 2026 regular_

### Official Review · Reviewer_y1xb · 2026-03-12

**Soundness:** 3
**Presentation:** 3
**Significance:** 2
**Originality:** 3
**Overall Recommendation:** 4
**Confidence:** 3

**Summary:**

This paper proposes Decoding Tree Sketching (DTS), a training-free decoding framework for large reasoning models. Instead of relying on independent multi-sample reasoning, DTS performs structured exploration by branching only at “decision tokens,” identified using low entropy and high varentropy, to build a compact decoding tree. It further applies an early-termination rule based on the observed negative correlation between reasoning length and accuracy, favoring earlier completed trajectories. Experiments on four models and four datasets show consistent improvements over standard inference, Self-Consistency, and DeepConf, while also reducing repetitive generation.

**Compliance With Llm Reviewing Policy:**

Affirmed.

**Final Justification:**

Most of my concerns are solved.

**Key Questions For Authors:**

- 1. The paper lacks an ablation on accuracy versus voting budget B. How does the marginal gain change as B increases from 1 to 8?
- 2.  Table 4 in Appendix C only performs the ablation under the AIME25 + DS-Distill-Qwen-1.5B setup. Are the thresholds $\tau_v=1.5, \tau_h=2.5$ optimal for all models/datasets?

**Limitations:**

yes

**Strengths And Weaknesses:**

**Strengths**
- 1.	The motivation is clear and the problem is well defined.
- 2.	The reduction in repetitive generation is significant. Table 2 shows that DTS-Greedy significantly reduces repetition, effectively addressing a key practical issue.
- 3.	The gains in inference efficiency are meaningful. Figure 5 shows that small models with DTS can match larger ones in performance, supporting a strong inference-time scaling argument.
- 4.	The method is simple and practical. The plug-and-play, training-free nature of the method also makes it appealing for real-world deployment.


**Weaknesses**
- 1.	Section 2.2 grounds its explanation in the GRPO training objective, and is essentially describing a preference induced by RL training. However, most of the experimental models in the paper are distilled reasoning models, such as DS-Distill-Qwen-1.5B, DS-Distill-Qwen-7B, and Phi4-mini-reasoning-4B. In other words, the paper uses an RL-based objective to explain the observed phenomenon, while the experiments do not sufficiently cover genuinely non-distilled RL-trained reasoning models.
- 2.	DTS branches into multiple new trajectories at decision tokens and maintains many active branches simultaneously. Compared with standard inference, how much additional GPU memory does this require in practice? The paper reports only FLOPs, but in real deployment, the memory cost of batched parallel branches may be the main bottleneck.
- 3.  Lack of validation on models larger than 10B.

---

> ### Author Rebuttal · Authors · 2026-03-30
>
> We thank the reviewer for the detailed questions.
>
> **[W1] The experiments do not sufficiently cover genuinely non-distilled RL-trained reasoning models.**
>
> **[ANS]** We thank the reviewer for raising this question. We clarify that our experiments already included two non-distilled reasoning models, Qwen3-0.6B and Phi-4-mini-reasoning-4B, where DTS shows consistent gains in Sec. 4.2. We do agree that broader coverage is valuable and have additionally evaluated the non-distilled Qwen3-14B in the table below. It is observed that DTS generally works on both distilled and non-distilled models. Given that GRPO introduces length bias into non-distilled models, and student models may inherit this bias from teacher models, DTS can potentially generalize well across both settings.
> |Model|Method|AIME24|AIME25|
> |-|-|-|-|
> |Qwen3-14B|Standard Inference|76.67|63.33|
> ||Self-Consistency|76.67|66.67|
> || DeepConf|80.00|73.33|
> ||DTS-Greedy|80.00|70.00|
> ||**DTS-Stable**|**86.67**|**76.67**|
> |Qwen3-0.6B|Standard Inference|11.33|14.00|
> ||Self-Consistency|11.33|25.33|
> ||DeepConf|13.33|24.00|
> ||DTS-Greedy|14.00|25.33|
> ||**DTS-Stable**|**18.67**|**29.33**|
> |Phi-4-mini-reasoning-4B|Standard Inference|49.33|36.00|
> ||Self-Consistency|61.33|42.00|
> ||DeepConf|62.67|38.00|
> ||DTS-Greedy|68.00|44.00|
> ||**DTS-Stable**|**71.33**|**52.00**|
>
> **[W2] How much additional GPU memory does this require in practice?**
>
> **[ANS]** We thank the reviewer for raising this question. We would like to respectfully clarify that, in practical scenarios, a key deployment consideration is the number of GPUs required to support inference. In this regard, DTS does not require additional GPUs compared to standard inference. It parallelizes branch exploration and reuses the prefix KV cache for efficient inference. As shown in the table below, DTS requires a comparable number of GPUs to standard inference, indicating that it does not introduce additional memory bottlenecks.
> |Model|Method|Acc.(%)|#A100|
> |-|-|-|-|
> |DS-Distill-Qwen-1.5B|Standard|26.67|1|
> ||DTS-Stable|**64.67**|1|
> |DS-Distill-Qwen|Standard|52.67|1|
> ||DTS-Stable|**78.67**|1|
> |Qwen3-14B|Standard|76.67|1|
> ||DTS-Stable|**86.67**|1|
>
> **[W3] Lack of validation on models larger than 10B.**
>
> **[ANS]** We thank the reviewer for this question. To address this, we have conducted additional experiments on Qwen3-14B and DS-Distill-Qwen-14B models on the AIME24 and AIME25 datasets in the table below. It is observed that DTS consistently outperforms baseline methods on larger models, indicating the its general effectiveness.
> |Model|Method|AIME24|AIME25|
> |-|-|-|-|
> |Qwen3-14B|Standard Inference|76.67|63.33|
> ||Self-Consistency|76.67|66.67|
> ||DeepConf|80.00|73.33|
> ||DTS-Greedy|80.00|70.00|
> ||**DTS-Stable**|**86.67**|**76.67**|
> |DS-Distill-Qwen-14B|Standard Inference|65.33|48.00|
> ||Self-Consistency|83.33|60.00|
> ||DeepConf|80.00|60.00|
> ||DTS-Greedy|80.00|63.33|
> ||**DTS-Stable**|**86.67**|**70.00**|
>
> **[KQ1] How does the marginal gain change as B increases from 1 to 8?**
>
> **[ANS]** We thank the reviewer for raising this question. We have conducted an additional ablation on DTS-Stable with voting budget $B \in {1,4,8\}$ across the four models and datasets (AIME24, AIME25, GPQA-D, LiveBench). The average accuracy over the four datasets is given in the table below. It is observed that the accuracy consistently improves as $B$ increases, which aligns with the design of DTS.
> |Model|B=1|B=4|B=8|
> |-|-|-|-|
> |DS-Distill-Qwen-1.5B|37.01|39.28|40.45|
> |DS-Distill-Qwen-7B|53.78|56.92|58.45|
> |Qwen3-0.6|23.37|25.88|27.32|
> |Phi-4-mini-reasoning-4B|54.48|56.02|58.09|
>
> **[KQ2] Are the thresholds $\tau_v=1.5, \tau_h=2.5$ optimal for all models/datasets?**
>
> **[ANS]** We thank the reviewer for raising this question. We would like to clarify that, as noted in Appendix A, all experiments use the same threshold $\tau_v=1.5, \tau_h=2.5$ and achieve consistent performance across different settings. As shown in the table below, DTS with $\tau_v=1.5, \tau_h=2.5$ consistently outperforms other settings, indicating its general effectiveness. In addition, performance remains stable across nearby values, indicating that DTS is robust and relatively insensitive to hyperparameter setting.
>
> | Model | Dataset | $\tau_v=1.5, \tau_h=1.2$ | $\tau_v=1.5, \tau_h=2.5$ | $\tau_v=1, \tau_h=2.5$ |
> |-|-|-|-|-|
> | DS-Distill-Qwen-1.5B | AIME24 | 64.00 | **64.67** | **64.67** |
> |         | AIME25 | 38.67 | **39.33** | **39.33** |
> | DS-Distill-Qwen-7B   | AIME24 | 76.67 | **78.67** | *78.00* |
> |         | AIME25 | **57.33** | *56.67* | 55.33 |

---

> > ### Author Rebuttal · Reviewer_y1xb · 2026-04-03
> >
> > Most of my concerns are solved.

---

> > > ### Author Response · Authors · 2026-04-03
> > >
> > > We sincerely thank the reviewer for acknowledging our rebuttal, and we are glad that we've resolved your concerns!

---

### Official Review · Reviewer_2LCP · 2026-03-12

**Soundness:** 3
**Presentation:** 3
**Significance:** 3
**Originality:** 3
**Overall Recommendation:** 4
**Confidence:** 4

**Summary:**

This paper proposes Decoding Tree Sketching (DTS), a training-free decoding framework for large reasoning models that explores multiple reasoning trajectories in a structured tree rather than by independent sampling. DTS branches only at "decision tokens" detected using entropy and varentropy of the next-token distribution, then selects outputs using an early-termination rule motivated by an observed anti-correlation between reasoning length and accuracy. The paper evaluates DTS-Greedy and DTS-Stable on four reasoning models and four datasets, and reports consistent gains over standard inference, Self-Consistency, and DeepConf, along with reduced repetitive generation.

**Compliance With Llm Reviewing Policy:**

Affirmed.

**Final Justification:**

My concerns have been addressed and I adjusted my score accordingly.

**Key Questions For Authors:**

1. For a fixed question, how often is the first branch to complete (DTS-Greedy's choice) actually the most accurate branch among all completed branches?
2. How does DTS compare against a simple shortest-trace baseline (select the shortest among N independently sampled completions) in terms of both accuracy and token cost?

**Limitations:**

The authors stated that "There are many potential societal consequences of our work, none which we feel must be specifically highlighted here." However, at least the authors are encouraged to discuss about the technique limitations and future work.

**Strengths And Weaknesses:**

**Strengths**
1. **Addresses a real and relevant failure mode.**
The repetition and overthinking problem in LRMs is well-motivated. Figure 6 provides an intuitive illustration of how DTS reaches a clean answer while standard inference exhausts the token budget.
2. **Consistent empirical gains across settings.**
Table 1 shows improvements over standard inference in all 16 model-dataset combinations, and DTS-Stable generally outperforms both Self-Consistency and DeepConf. The overall trend is difficult to dismiss.
3. **Clear presentation.**
The method is specified clearly enough in Equations (2)–(4) and Algorithm 1.

**Weaknesses**
1. **Limited novelty relative to prior work.**
The core mechanism of DTS, branching at tokens where entropy and varentropy jointly exceed thresholds, is very similar to several existing works:
(1) Entropix (xjdr-alt, 2024) proposes the same entropy/varentropy-triggered branching at the system level;
(2) Entropy-Aware Branching (Li et al., arXiv 2503.21961) applies an essentially identical selective branching criterion to math reasoning; (3) CoT Decoding (Wang & Zhou, NeurIPS 2024) establishes top-K token branching during decoding as a general principle. The early-termination heuristic is the most distinct element of DTS, but the paper does not frame or evaluate it as the primary contribution.
2. **The selection principle is justified by aggregate correlation but applied as an instance-level rule.**
Section 2.2 and Figure 2 show that, in aggregate, longer traces are less accurate. DTS-Greedy then turns this into a per-instance decision: stop at the first completed branch. This is a much stronger claim than the evidence supports. A per-prompt analysis, how often is the earliest-completed branch the best branch for the same question, is necessary to validate this core inductive leap.
3. **The branching trigger in Equation (2) is central but almost unvalidated.**
No ablations compare entropy-only, varentropy-only, random branching, or unconditional top-K branching. Without these, it is impossible to determine whether the joint entropy/varentropy criterion provides meaningful benefit over simpler alternatives, or whether the detected positions actually correspond to semantically meaningful reasoning divergences.
4. **A critical baseline is missing.**
The paper does not include a shortest-trace baseline: sample N independent trajectories and select the shortest completed one. Without this control, it is unclear whether gains stem from structured tree exploration or simply from the preference for short outputs. This is arguably the most important missing comparison given the paper's causal story.

---

> ### Author Rebuttal · Authors · 2026-03-30
>
> We thank the reviewer for the detailed questions.
>
> **[W1] Limited novelty relative to prior work.**
>
> **[ANS]** We respectfully disagree with the assessment of limited novelty, and would like to clarify that DTS is significantly different from the cited works:
>
> **(1) Entropix (xjdr-alt, 2024):** First, this GitHub repository **does not provide a concrete algorithm for structured tree construction**. Their repository explicitly notes that "*# TODO(xjdr): We need to do a differnt version of tree search here*" in their empty branching function. Thus, structured reasoning tree construction, which is central to DTS, is not realized. Second, Extropix **does not include any trajectory selection mechanism**, another core component of DTS. Third, Extropix targets base LLMs, whereas DTS is for reasoning models and addresses their unique behaviors (e.g., repetition, length bias).
>
> **(2) Entropy-Gated Branching (EGB):** This work differs from DTS in two key aspects. First, it relies on uncertainty high entropy branching, whereas DTS proposes **semantics branching** through high varentropy + low entropy. Second, **EGB relies on an external reward model** for trajectory evaluation and selection, incurring extra compute, while DTS is fully plug-and-play and uses only internal model signals, making it more practical. We note this work is contemporaneous (accepted recently in January).
>
> **(3) CoT Decoding:** This work differs from DTS in both exploration and selection. It performs near-independent sampling since it **only branches over top-K tokens at the first decoding step**. DTS instead performs semantics branching throughout decoding, enabling structured exploration of the reasoning space. For selection, it huristically relies on confidence scoring, while DTS proposes to select the short yet reliable trajectory that is empirically grounded by the anti-correlation property of LRMs.
>
> To further clarify, we summarize key differences below:
> |Method|Structured Exploration|Semantics Branching|Plug-and-Play|Rule-based Selection|
> |-|-|-|-|-|
> |Entropix|✗|✓|✗|✗|
> |CoT Decoding|✗|✗|✓|✓|
> |EGB|✓|✗|✗|✗|
> |**DTS**|**✓**|**✓**|**✓**|**✓**|**✓**|
>
> We respectfully believe that our work demonstrates clear novelty compared with existing work, as also acknowledged by Reviewer `#eWYw` that _interesting and practical inference-time idea with training-free, model agnostic, and easy to position_.
>
>
> **[W2, KQ1] The selection principle is justified by aggregate correlation but applied as an instance-level rule.**
>
> **[ANS]** To address this, we have conducted a within-question (macro) analysis that normalizes accuracy per prompt before averaging, debiasing the effect of difficulty. We report results in the table below, where each value is the macro-averaged accuracy(%) within a length bin, demonstrating that our **length-accuracy anti-correlation persists at instance level**.
> |Model|Dataset|2×10³|4×10³|8×10³|12×10³|
> |-|-|-|-|-|-|
> |DS-1.5B|AIME24|**79.2**|61.4|49.7|22.1|
> ||AIME25|**83.3**|54.4|31.1|20.5|
> |DS-7B|AIME24|**98.3**|84.7|68.8|46.5|
> ||AIME25|**83.3**|68.6|47.2|31.1|
>
> Regarding KQ1, we additionally compare the probability that the shortest trajectory (DTS-Greedy) is the most accurate with the probability that a randomly selected trajectory is the most accurate. We provide the probabilities in the table below, demonstrating that selecting the shortest trajectory has a higher than 80% chance on average, which further validates DTS's selection principle.
> |Model|Dataset|DTS-Greedy P(%)|Random P(%)|
> |-|-|-|-|
> |DS-7B|AIME25|**90.8**|36.0|
> ||GPQA-D|**85.4**|49.3|
> |Phi-4-4B|AIME25|**81.7**|36.0|
> ||GPQA-D|**80.8**|50.2|
>
> **[W3] The branching trigger in Equation (2) is central but almost unvalidated.**
>
> **[ANS]** We respectfully disagree with this point. **The evaluation of branching trigger has been explicitly given in Table 4.** Following the reviewer’s suggestion, we conduct extra ablations with a broader set of triggers, as shown below. It is observed that DTS consistently achieves the best performance under our proposed criterion, indicating its effectiveness.
> |Criteria|DTS-Greedy|DTS-Stable|
> |-|-|-|
> |$H(P_t)\ge\tau_h$|29.3|33.3|
> |$VE(P_t)\ge\tau_v\text{ and }H(P_t)\le\tau_h$|**34.7**|**39.3**|
> |$VE(P_t)\ge\tau_v$|34.0|38.7|
> |Random|28.7|33.3|
>
> **[W4, KQ2] A critical baseline is missing.**
>
> **[ANS]** We include the suggested baseline that applies independent sampling and shortest trajectory selection in the table below (ACC/Length). The results show that DTS outperforms both 'w/o exploration' and 'w/o shortest & exploration', indicating that both structured exploration and shortest trajectory selection significantly contribute to the improvement.
> |Model|Dataset|DTS-Greedy|w/o exploration|w/o shortest & exploration|
> |-|-|-|-|-|
> |DS-1.5B|AIME24|**54.7/5174**|*43.3/5637*|26.7/16596|
> ||AIME25|**34.7/6232**|*31.3/6772*|24.7/17809|
> |DS-7B|AIME24|**73.3/6184**|*66.7/6870*|52.7/13902|
> ||AIME25|**53.3/6724**|*52.7/7357*|36.0/15053|

---

> > ### Author Rebuttal · Reviewer_2LCP · 2026-04-01
> >
> > I thank the authors for the thorough rebuttal. The responses to W2–W4 are convincing. I have adjusted our score accordingly.
> >
> > However, myconcern on W1 (novelty) is not fully resolved. For me, it seems like that DTS can be decomposed as **A (entropy/varentropy-based branching) + B (shortest-trace selection)**, where both components have close precedents:
> >
> > - **Component A**: EGB [Li et al., arXiv 2503.21961] performs entropy-gated branching to construct reasoning trees; ETTRL [arXiv 2508.11356] branches at highest-entropy tokens; TreeRL [ACL 2025] uses entropy-based forking with ablations against random branching.
> > - **Component B**: Devic et al. [1] systematically establish that trace length is a reliable uncertainty signal in reasoning models, directly motivating shortest-trace selection.
> >
> > [1] Trace Length is a Simple Uncertainty Signal in Reasoning Models arXiv 2510.10409
> >
> > I acknowledge that combining known components can yield meaningful contributions. However, to establish that DTS's specific integration provides value beyond a naive assembly, I would find it informative to see a comparison against **EGB branching + shortest-trace selection** (i.e., replacing DTS's varentropy+low-entropy criterion with EGB's high-entropy criterion, while keeping the same selection rule). This would isolate whether the proposed branching criterion is the critical ingredient or whether most gains come from the combination itself. I am willing to further adjust our score if this comparison is provided.

---

> > > ### Author Response · Authors · 2026-04-02
> > >
> > > We thank the reviewer for the follow-up questions.
> > >
> > > **[Q1] Direct comparison with EGB branching + shortest-trace selection.**
> > >
> > > **[ANS]** We sincerely value this suggestion and conduct additional experiments comparing DTS with EGB + shortest-trace selection. The results are shown in the table below. It is observed that DTS consistently achieves higher accuracy than EGB + shortest-trace selection. This indicates the effectiveness of DTS’s tree-structured exploration and semantic branching strategy.
> > >
> > > |Model|Dataset|DTS-Stable|EGB+shortest-trace selection|
> > > |-|-|-|-|
> > > |DS-Distill-Qwen-1.5B|AIME24|**64.67**|45.33|
> > > ||AIME25|**39.33**|33.33|
> > > |DS-Distill-Qwen-7B|AIME24|**78.67**|70.00|
> > > ||AIME25|**56.67**|50.67|
> > > |Qwen3-0.6B|AIME24|**18.67**|15.33|
> > > ||AIME25|**29.33**|26.00|
> > > |Phi4-4B|AIME24|**71.33**|64.67|
> > > ||AIME25|**52.00**|42.67|
> > >
> > > **[Q2] Novelty compared to related works.**
> > >
> > > **[ANS]** We would like to respectfully clarify that TreeRL [1], ETTRL [2], and Devic et al. [3] do not undermine the contribution of DTS, as they are fundamentally orthogonal to DTS. We elaborate below:
> > >
> > > - First, TreeRL [1] and ETTRL [2] aim to enhance exploration **during the RL training**. In contrast, DTS is a **training-free** method that operates purely at inference time. This distinction makes DTS a plug-and-play solution that can be readily applied to existing LRMs without training, thereby providing strong complementarity to training-based methods.
> > >
> > > - Second, Devic et al. [3] differ substantially from DTS, because it focuses on **diagnostic and analytical understanding of LRMs**. In contrast, DTS focuses on developing a **practical and deployable framework** to improve LRM reasoning. To this end, DTS develops a tree-structured exploration mechanism with semantic branching conditions, along with a strategy of selecting short yet reliable trajectories. Therefore, DTS is conceptually and methodologically orthogonal to Devic et al. [3].
> > >
> > > We thank the reviewer again for these questions and hope these additional evidences and clarifications address the reviewer's concern.
> > >
> > > [1] "Treerl: Llm reinforcement learning with on-policy tree search." ACL 2025.
> > >
> > > [2] "Ettrl: Balancing exploration and exploitation in llm test-time reinforcement learning via entropy mechanism." arXiv.
> > >
> > > [3] "Trace length is a simple uncertainty signal in reasoning models." arXiv.

---

### Official Review · Reviewer_FaCY · 2026-03-13

**Soundness:** 3
**Presentation:** 4
**Significance:** 2
**Originality:** 3
**Overall Recommendation:** 4
**Confidence:** 4

**Summary:**

This paper proposes Decoding Tree Sketching (DTS), a training-free decoding framework aimed at improving the reasoning performance of large reasoning models (LRMs). The method treats the reasoning process as a search over an implicit token-level reasoning tree. Instead of generating independent reasoning trajectories as in self-consistency, DTS selectively branches at decision tokens, which are identified based on entropy and varentropy of the next-token distribution. These branches are expanded in parallel to explore multiple reasoning paths.

A central concept presented by the paper is that reasoning trajectories generated during stochastic decoding can be organized as a tree structure, and structured exploration of this space may uncover better solutions than independent sampling. The authors seek to present a pertinent issue in current parallel reasoning methods: existing approaches often generate redundant trajectories and only perform selection after all trajectories complete, which may lead to inefficient exploration.

To address this, DTS introduces two main components: (1) selective branching at decision tokens to construct a compact reasoning tree, and (2) an early termination strategy motivated by an empirical observation that reasoning length negatively correlates with accuracy. The earliest completed trajectory is therefore prioritized as the final answer.

Experiments on four reasoning models across four reasoning benchmarks show consistent accuracy improvements compared to standard inference and several multi-trajectory baselines.

**Compliance With Llm Reviewing Policy:**

Affirmed.

**Key Questions For Authors:**

see weakness above

**Limitations:**

yes

**Strengths And Weaknesses:**

- Strengths

1. The method is simple and practical

The strongest aspect of the paper is its simplicity. DTS is a decoding-time method, requires no retraining, and can be applied to existing reasoning models directly. This makes the work easy to understand and potentially useful in practice.

2. The experimental narrative is clean and logically presented

The paper is well organized. The motivation, method, and experiments are aligned around a single hypothesis, making the overall story easy to follow. The empirical results are presented clearly, and the paper does a good job of connecting the algorithmic design to the experimental observations.

- Weaknesses

1. The trajectory-selection claim is not fully validated

The main claim of the paper is not that shorter reasoning is universally better, but that selecting the shortest completed trajectory is a better selection heuristic in parallel decoding. However, the paper does not sufficiently validate this claim experimentally.

For example, when multiple candidate trajectories are available, it would be important to compare selection strategies such as:
	•	shortest trajectory
	•	random trajectory
	•	random-length trajectory
	•	majority voting
	•	confidence-based selection

Without such ablations, it is difficult to determine whether the observed improvements truly come from the proposed shortest-trajectory heuristic, or simply from increased trajectory exploration.

2. Dependence on a model-specific phenomenon

The proposed method is motivated by an empirical length–accuracy anti-correlation observed in current RL-trained reasoning models. However, this phenomenon may not generalize broadly. In many reasoning tasks, especially complex mathematical or algorithmic problems, correct solutions often require longer reasoning chains.

Moreover, recent training approaches increasingly attempt to reduce length bias and encourage adaptive reasoning depth. If future reasoning models exhibit less of this bias, the advantage of the proposed strategy may diminish.

3. Experimental baselines could be stronger

The evaluation primarily compares DTS with standard inference, self-consistency, and DeepConf. However, stronger or more standard stochastic decoding strategies for reasoning models—such as top-p / top-k sampling settings or other search-based reasoning approaches—are not included. Adding these baselines would provide a clearer understanding of where the gains originate.

---

> ### Author Rebuttal · Authors · 2026-03-30
>
> We thank the reviewer for the detailed questions.
>
> **[W1] The trajectory-selection claim is not fully validated.**
>
> **[ANS]** We would like to clarify that Sec. 4.4 (Table 3) already includes an ablation comparing shortest trajectory selection with max-logit confidence. We sincerely value this comment and conduct more experiments of accuracy(%) comparison in the table below on four models and datasets, using shortest, random, and confidence selection. We show that shortest selection achieves better performance than others.
> |Model|Dataset|Shortest|Confidence|Random|
> |-|-|-|-|-|
> |DS-Distill-Qwen-1.5B|AIME24|**54.67**|38.67|26.67|
> ||AIME25|**34.67**|21.33|24.67|
> ||GPQA-D|**41.41**|39.39|32.02|
> ||LiveBench|**17.30**|13.10|6.00|
> |DS-Distill-Qwen-7B|AIME24|**73.33**|65.33|52.67|
> ||AIME25|**53.33**|50.00|36.00|
> ||GPQA-D|**55.76**|53.94|49.29|
> ||LiveBench|**32.70**|30.30|27.20|
> |Qwen3-0.6B|AIME24|**14.00**|7.33|11.33|
> ||AIME25|**25.33**|16.67|14.00|
> ||GPQA-D|**25.56**|24.75|24.75|
> ||LiveBench|**28.60**|21.80|24.70|
> |Phi-4-mini-reasoning-4B|AIME24|**68.00**|58.00|49.33|
> ||AIME25|**44.00**|42.00|36.00|
> ||GPQA-D|**53.13**|53.03|50.20|
> ||LiveBench|**52.80**|50.40|42.40|
>
> We provide additional albation by disabling exploration (independent sampling) and shortest trajectory selection modules and compare accuracy(%) below, demonstrating both modules contribute to the performance gains.
> |Model|Dataset|DTS-Greedy|shortest w/o exploration|w/o shortest w/o exploration|
> |-|-|-|-|-|
> |DS-Distill-Qwen-1.5B|AIME24|**54.67**|*43.33*|26.67|
> ||AIME25|**34.67**|*31.33*|24.67|
> |DS-Distill-Qwen-7B|AIME24|**73.33**|*66.67*|52.67|
> ||AIME25|**53.33**|*52.67*|36.00|
>
> **[W2.1] Dependence on a model-specific phenomenon.**
>
> **[ANS]** We thank the reviewer for this question. We respectfully disagree that DTS relies on a non-generalizable phenomenon, and would like to clarify this from two aspects.
>
> - First, the length-accuracy anti-correlation is not model- or task-specific, but arises from the GRPO training objective, which introduces a systematic length bias independent of downstream tasks, as discussed in Sec. 2.2. In Sec. 4.2, we already show that DTS generalizes across multiple domains, including AIME24/25, two of the hardest public math benchmarks, and Livebench, which contains algorithmic and logical tasks such as SAT-style reasoning and Zebra puzzles.
>
> - Second, while correct reasoning requires completeness, incorrect reasoning often leads to redundant or repetitive steps and results in longer trajectories, as discussed in Sec. 4.3, 4.6, and Appendix B. Empirically, we report a **within-question (macro) accuracy that debias the influence of question difficulty**. Specifically, macro accuracy is given by $Acc_{macro}(b)=\frac{1}{N}\sum_{i=1}^{N}\frac{\sum_{j\in b}\mathbb{I}[correct_{i,j}]}{|\{j\in b\}|}$, where $b$ denote a length bin and $N$ is the number of questions. The length normalization can debias the influence of question difficulty on reasoning length. We report results in the table below, where each value is the macro-averaged accuracy(%) within a length bin. It is observed that the length–accuracy anti-correlation holds significantly without the influence of question difficulty and is not task-specific, indicating a consistent behavioral pattern in LRMs.
> |Model|Dataset|2×10³|4×10³|8×10³|12×10³|
> |-|-|-|-|-|-|
> |DS-Distill-Qwen-1.5B|AIME24|**79.2**|61.4|49.7|22.1|
> ||AIME25|**83.3**|54.4|31.1|20.5|
> |DS-Distill-Qwen-7B|AIME24|**98.3**|84.7|68.8|46.5|
> ||AIME25|**83.3**|68.6|47.2|31.1|
>
> **[W2.2] Recent training approaches increasingly attempt to reduce length bias.**
>
> **[ANS]** We agree that mitigating GRPO's length bias is an important direction. However, current evidence suggests this issue remains unresolved. For example, one of the most impactful work, Dr.GRPO, cannot fully address this issue [1]. To further validate this, we evaluate two Dr.GRPO-trained LRMs below. It is observed that DTS consistently improves over standard inference and Self-Consistency. Therefore, we believe DTS and the length-accuracy anti-correlation remain relevant under the current state of the field.
> |Model|Dataset|Std. Inf.|+SC|+DTS|
> |-|-|-|-|-|
> |Qwen2.5-Math-1.5B+Dr.GRPO|AIME24|15.33|17.33|**19.33**|
> ||AIME25|10.00|14.67|**16.67**|
> |Qwen2.5-Math-7B+Dr.GRPO|AIME24|31.33|38.00|**39.33**|
> ||AIME25|11.33|16.67|**18.67**|
>
> [1] "Trace length is a simple uncertainty signal in reasoning models." arXiv.
>
> **[W3] Experimental baselines could be stronger.**
>
> **[ANS]** We thank the reviewer for this question, and added two more baseline methods, inference with top-k and top-p sampling [2]. The following table shows that DTS outperforms top-k and top-p sampling, indicating its effectiveness.
> |Model|Dataset|DTS-Greedy|top-k|top-p|
> |-|-|-|-|-|
> |DS-Distill-Qwen-1.5B|AIME24|**54.67**|43.33|34.67|
> ||AIME25|**34.67**|29.33|23.33|
> |DS-Distill-Qwen-7B|AIME24|**73.33**|69.33|66.67|
> ||AIME25|**53.33**|43.33|42.67|
>
> [2]"Chain-of-thought reasoning without prompting." NeurIPS.

---

> > ### Author Rebuttal · Reviewer_FaCY · 2026-04-04
> >
> > Thank you to the authors for their response. My concerns have been addressed. Since I have already given a positive score of 4, I will maintain my current rating.

---

> > > ### Author Response · Authors · 2026-04-04
> > >
> > > We sincerely thank the reviewer for the positive recognition, and we are glad that we've addressed your concerns!

---

### Official Review · Reviewer_eWYw · 2026-03-14

**Soundness:** 3
**Presentation:** 3
**Significance:** 3
**Originality:** 3
**Overall Recommendation:** 4
**Confidence:** 4

**Summary:**

The paper proposes Decoding Tree Sketching (DTS). It's a training-free inference-time decoding method for LLM reasoning. The core idea is to represent the space of possible reasoning trajectories as a tree, branch only at selected “decision tokens,” and terminate early in favor of shorter completed trajectories. Decision tokens are detected using a low entropy + high varentropy criterion on the next token distribution. Across four models and four reasoning benchmarks, the paper reports consistent accuracy gains over standard inference, Self Consistency, and DeepConf, together with lower repetition rates.

**Compliance With Llm Reviewing Policy:**

Affirmed.

**Key Questions For Authors:**

See the weakness

**Limitations:**

No, the paper doesn't discuss limitations. Currently, the evaluation is limited to relatively small distilled reasoning models, so it is unclear whether the method transfers to stronger or larger models.

**Strengths And Weaknesses:**

**Strengths**

- Interesting and practical inference-time idea. The method is training-free, model agnostic in scope, and easy to position relative to existing parallel thinking methods.

- Empirical gains are fairly broad within the paper’s setup. DTS Greedy and DTS Stable both outperform standard inference and usually beat Self Consistency and DeepConf across AIME24, AIME25, GPQA Diamond, and LiveBench for four LRMs.

- The paper is clearly written and easy to follow. The main ideas are presented in a straightforward way, the method is decomposed cleanly into selective branching and early termination, and the overall narrative is easy to understand even when the underlying motivation is partly heuristic.

**Weakness**:
- The evaluation is limited to relatively small distilled reasoning models, leaving it unclear whether the proposed branching and shortest path heuristics continue to help for substantially stronger or larger-scale models. Since both core assumptions of the method appear tied to the statistical behavior of current post-trained LRMs, the paper would be stronger with evidence that DTS scales beyond these small models.
- The reported length accuracy anti-correlation is interesting, but a key confounder seems unaddressed: harder questions may naturally induce longer reasoning traces and lower success rates, which could produce the observed negative correlation even if shorter reasoning is not intrinsically better. The claim would be much stronger with within-question analyses or other causal analysis tools.

---

> ### Author Rebuttal · Authors · 2026-03-30
>
> We thank the reviewer for the detailed questions.
>
> **[W1] The evaluation is limited to relatively small distilled reasoning models.**
>
> **[ANS]** We thank the reviewer for raising this question. To address this, we have conducted additional experiments on Phi-4-mini-reasoning (non-distilled), Qwen3-14B (non-distilled, large), and DS-Distill-Qwen-14B (large) to show the scalability of DTS to bigger and non-distilled models. The accuracy(%) results are given in the following table, where DTS outperforms baseline methods, demonstrating the effectiveness on LRMs at different scales.
> |Model|Method|AIME24|AIME25|
> |-|-|-|-|
> |Phi-4-mini-reasoning-4B|Standard Inference|49.33|36.00|
> ||Self-Consistency|61.33|42.00|
> ||DeepConf|62.67|38.00|
> ||DTS-Greedy|68.00|44.00|
> ||**DTS-Stable**|**71.33**|**52.00**|
> |Qwen3-14B|Standard Inference|76.67|63.33|
> ||Self-Consistency|76.67|66.67|
> ||DeepConf|80.00|73.33|
> ||DTS-Greedy|80.00|70.00|
> ||**DTS-Stable**|**86.67**|**76.67**|
> |DS-Distill-Qwen-14B|Standard Inference|65.33|48.00|
> ||Self-Consistency|83.33|60.00|
> ||DeepConf|80.00|60.00|
> ||DTS-Greedy|80.00|63.33|
> ||**DTS-Stable**|**86.67**|**70.00**|
>
> **[W2] Harder questions may naturally induce longer reasoning traces and lower success rates**
>
> **[ANS]** We thank the reviewer for raising this question. To address this, we report a **within-question (macro) accuracy that debiases the influence of question difficulty**. Specifically, macro accuracy is estimated by $Acc_{macro}(b)=\frac{1}{N}\sum_{i=1}^{N}\frac{\sum_{j\in b}\mathbb{I}[correct_{i,j}]}{|\{j\in b\}|}$, where $b$ denote a length bin and $N$ is the number of questions. The length normalization can debias the influence of question difficulty on reasoning length. We report results in the following table, where each value is the macro-averaged accuracy(%) within a length bin. It is observed that the accuracy consistently decreases as reasoning length grows, indicating the length–accuracy anti-correlation without the influence of question difficulty.
> |Model|Dataset|2×10³|4×10³|8×10³|12×10³|
> |-|-|-|-|-|-|
> |DS-Distill-Qwen-1.5B|AIME24|**79.2**|61.4|49.7|22.1|
> ||AIME25|**83.3**|54.4|31.1|20.5|
> |DS-Distill-Qwen-7B|AIME24|**98.3**|84.7|68.8|46.5|
> ||AIME25|**83.3**|68.6|47.2|31.1|

---

> > ### Author Rebuttal · Reviewer_eWYw · 2026-04-05
> >
> > Thanks for the clarification and additional experiments. Two minor comments: I don't think Phi-4-mini-reasoning and Qwen3- 14B are "not distilled"; The results on AIME24/AIME25 without testing avg@n can have very large variance. I'd keep my positive score.

---

> > > ### Author Response · Authors · 2026-04-06
> > >
> > > We thank the reviewer for the follow-up questions.
> > >
> > > **[Q1] More experiments on non-distilled models.**
> > >
> > > **[ANS]** We thank the reviewer for raising this question. To address this, we conduct additional experiments on Olmo-3.1-7B-RL-Zero-Math, a non-distilled model built upon GRPO without post-training distillation from larger models according to their report [1]. We report avg@5 accuracy(%) below and demonstrate that DTS consistently outperforms baseline methods, showing its effectiveness beyond distilled models.
> > >
> > > |Model|Dataset|Standard Inference|Self-Consistency|DTS-Stable|
> > > |-|-|-|-|-|
> > > |Olmo-3.1-7B-RL-Zero-Math|AIME24|50.67|61.33|**72.00**|
> > > ||AIME25|38.00|46.67|**55.33**|
> > >
> > > **[Q2] Testing avg@n accuracy.**
> > >
> > > **[ANS]** We thank the reviewer for raising this question. To address this, we conducted additional repeated experiments on DS-Distill-Qwen-14B and report avg@5 accuracy(%). As shown in the table below, DTS maintains consistent and stable gains over baselines across repeated runs on larger models. We would also like to respectfully clarify that all results in the main paper are reported using avg@5 accuracy, as specified in Appendix A.
> > >
> > > |Model|Method|AIME24|AIME25|
> > > |-|-|-|-|
> > > |DS-Distill-Qwen-14B|Standard Inference|65.33|48.00|
> > > ||Self-Consistency|82.67|60.67|
> > > ||DeepConf|81.33|62.00|
> > > ||DTS-Greedy|82.00|63.33|
> > > ||**DTS-Stable**|**87.33**|**69.33**|
> > >
> > > We thank the reviewer again for these questions and hope these additional evidences and clarifications address the reviewer's concern.
> > >
> > > [1] Olmo, Team, et al. "Olmo 3." arXiv.

---

### Decision · Program_Chairs · 2026-04-30

**Decision:**

Accept (regular)

**Comment:**

This paper proposed Decoding Tree Sketching (DTS), a novel plug-and-play decoding framework to improve LLM reasoning in multi-trajectory exploration and reasoning selection. Before rebuttal stage, all reviewers agree with the practicality of the proposed method, but also point out some issues on experimental results, generalization and efficiency. During the rebuttal stage, authors have provided thorough experiments and responses to address these concerns. All reviewers lean to a positive opinion to this paper. Therefore, I recommend acceptance for this paper.